# Asynchronous Antarctic and Greenland ice-volume contributions to the last interglacial sea-level highstand

Eelco J. Rohling [1,2,7]*, Fiona D. Hibbert [1,7]*, Katharine M. Grant[1], Eirik V. Galaasen [3], Nil Irvalı [3], Helga F. Kleiven [3], Gianluca Marino[1,4], Ulysses Ninnemann[3], Andrew P. Roberts[1], Yair Rosenthal[5], Hartmut Schulz[6], Felicity H. Williams [1] & Jimin Yu [1]

The last interglacial (LIG; ~130 to ~118 thousand years ago, ka) was the last time global sea level rose well above the present level. Greenland Ice Sheet (GrIS) contributions were insufficient to explain the highstand, so that substantial Antarctic Ice Sheet (AIS) reduction is implied. However, the nature and drivers of GrIS and AIS reductions remain enigmatic, even though they may be critical for understanding future sea-level rise. Here we complement existing records with new data, and reveal that the LIG contained an AIS-derived highstand from ~129.5 to ~125 ka, a lowstand centred on 125–124 ka, and joint AIS + GrIS contributions from ~123.5 to ~118 ka. Moreover, a dual substructure within the first highstand suggests temporal variability in the AIS contributions. Implied rates of sea-level rise are high (up to several meters per century; m c$^{-1}$), and lend credibility to high rates inferred by ice modelling under certain ice-shelf instability parameterisations.

[1] Research School of Earth Sciences, The Australian National University, Canberra, ACT 2601, Australia. [2] Ocean and Earth Science, University of Southampton, National Oceanography Centre, Southampton SO14 3ZH, UK. [3] Department of Earth Science and Bjerknes Centre for Climate Research, University of Bergen, Allegaten 41, 5007 Bergen, Norway. [4] Department of Marine Geosciences and Territorial Planning, University of Vigo, 36310 Vigo, Spain. [5] Institute of Marine and Coastal Sciences, Rutgers University, New Brunswick, NJ 08903, USA. [6] Department of Geology and Paleontology, University of Tuebingen, Sigwartstrasse 10, D-7400 Tuebingen, Germany. [7] These authors contributed equally: Eelco J. Rohling, Fiona D. Hibbert. *email: eelco.rohling@anu.edu.au; fiona.hibbert@anu.edu.au

The magnitudes and rates of mass reductions in today's remaining ice sheets (GrIS and AIS) in response to (past or future) warming beyond pre-industrial levels remain poorly understood. With sea levels reaching a highstand of +6 to +9 m[1–3], or up to 2 m higher[4], relative to the present (hereafter 0 m), the last interglacial (LIG) is a critical test-bed for improving this understanding. Thermosteric and mountain glacier contributions fell within 0.4 ± 0.3 m and at most 0.3 ± 0.1 m, respectively[5,6], and also Greenland Ice Sheet (GrIS) contributions were insufficient to explain the LIG highstand[7–9]. Hence, substantial Antarctic Ice Sheet (AIS) reduction is implied[1–3]. Determining AIS and GrIS sea-level contributions during the LIG in more detail requires detailed records with tightly constrained chronologies, along with statistical and model-driven assessments (e.g., see refs. [1–3,9–15]; Supplementary Note 1). To date, however, chronological (both absolute and relative) and/or vertical uncertainties in LIG sea-level data have obscured details of the timings, rates, and origins of change.

Age control is most precise for radiometrically dated coral-based sea-level data, but stratigraphically discontinuous LIG coverage of these complex three-dimensional systems, and species- or region-specific habitat-depth uncertainties affect the inferred sea-level estimates[11]. Stratigraphic coherence and, therefore, relative age relationships among samples are stronger in the sediment-core-based Red Sea relative sea-level (RSL) record[1,10,16–18] (Methods), but its LIG signals initially lacked replication and sufficient age control[1,17]. Chronological alignment of the Red Sea record with radiometrically dated speleothem records has since settled its age for the LIG-onset[10,18,19], but the LIG-end remains poorly constrained (Methods). Also, the Red Sea record has since 2008 (ref. [1]) been a statistical stack of several records without the tight sample-to-sample stratigraphy of contiguous sampling through a single core, and this has obscured details that are essential for studying centennial-scale changes[10,17–19]. Advances in understanding LIG sea-level contributions therefore relied on statistical deconvolutions based on multiple datasets and associated evaluations with ambiguous combining of chronologies[2,12,13,20], or considered only mean LIG contributions[21]. Some of these studies suggest that AIS contributions likely preceded GrIS contributions, and that there were intra-LIG sea-level fluctuations, with kilo-year averaged rates of at most 1.1 m per century (and likely smaller)[13], though this does not discount higher values for centennial-scale averages (e.g., ref. [1]).

To quantify centennial-scale average sea-level-rate estimates that may reveal rapid events and processes of relevance to the future, and robustly distinguish AIS from GrIS contributions, we present an approach that integrates precise event-dating from coral/reef and speleothem records[3,22–24] with stratigraphically tightly constrained Red Sea sea-level records and a broad suite of palaeoceanographic evidence. Results indicate that the LIG contained an early AIS-derived highstand, followed by a drop centred on 125–124 ka, and then joint AIS + GrIS contributions for the remainder of the LIG. We also infer high rates of sea-level change (up to several metres per century; m c$^{-1}$), that likely reflect complex interactions between oceanic warming, dynamic ice-mass loss, and glacio-isostatic responses.

## Results

### Overview of LIG sea-level evidence.
The nature of LIG sea-level variability remains strongly debated, with emphasis on two issues. First, near-field sites (close to the ice sheets) in NW Europe suggest LIG sea-level stability, although resolution and age control remain limited and other N European sites might support sea-level fluctuations[25]. Second, there is a wealth of global sites (mostly in the far field relative to the ice sheets) that implies LIG

sea-level variability (Fig. 1), but which also reveals a striking divergence between site-specific signals with respect to both timing and amplitude of variability (Supplementary Note 1). This suggests that individual sites are overprinted by considerable site-specific influences—e.g., prevailing isostatic, tectonic, physical, biological, biophysical, and biochemical characteristics—rather than reflecting only global sea-level changes. Regardless, a more coherent pattern seems to be emerging from the more densely dated and stratigraphically well-constrained sites, which include the Seychelles, Bahamas, and also Western Australia (Supplementary Note 1, synthesis). The Seychelles coral data are radiometrically precisely dated, avoid glacio-isostatic offsets among sites, and include stratigraphic relationships that unambiguously reveal relative event timings[3,22]. The Bahamas data comprise stratigraphically well-documented and dated evidence of different reef-growth phases[23]. Nevertheless, the overall coral-based literature suggests at least two plausible types of LIG history (early vs. late highstand solutions) that remain to be reconciled (Supplementary Note 1, synthesis).

### Updated Red Sea age model.
Regarding the Red Sea RSL record, we improve its LIG-end age control[10,18] by comparing the entire dataset (the stack) with radiometrically dated coral-data compilations[11,26] and Yucatan cave-deposits that indicate when sea level dropped below the cave (i.e., a "ceiling" for sea level)[24]. This comparison reveals that the 95% probability limit of the Red Sea stack on its latest chronology[10,19] dropped too early (123 ka; see Methods and Supplementary Note 2) relative to the well-dated archives (119–118 ka; Fig. 2b, c; Supplementary Figs. 2 and 3). We, therefore, adjust this point to 118.5 ± 1.2 ka (95% uncertainty bounds) (Fig. 2, Supplementary Figs. 2 and 3), and accordingly revise all interpolated LIG ages with fully propagated uncertainties (Supplementary Fig. 2).

### Estimates of Greenland mass loss.
Next, we compare the Red Sea sea-level information (Fig. 2b, c, e, f) with estimates of GrIS-derived LIG sea-level contributions from a model-data-assimilation of Greenland ice-core data for summer temperature anomalies, accumulation rates, and elevation changes[9] (Fig. 2a). We add independent support for the inferred late GrIS contribution[9], based on a newly extended record of sea-water oxygen isotope ratios ($\delta^{18}O_{sw}$) from a sediment core from Eirik Drift, off southern Greenland. In this location, $\delta^{18}O_{sw}$ reflects Greenland meltwater input with a sensitivity of 4 ± 1.2 m global sea-level rise for the −1.3‰ change seen in the $\delta^{18}O_{sw}$ record from ~128 to ~118 ka (Fig. 2a) (Methods, Supplementary Note 3). This record suggests (albeit within combined uncertainties) generally lower GrIS contributions than Yau et al.[9], which may agree with results from other modelling studies for GrIS[14,15]. Both the modelling and $\delta^{18}O_{sw}$ approaches indicate a late GrIS contribution to LIG sea level, which is further supported by wider N. Atlantic and European palaeoclimate data, which reveal that contributions started after 127 ka, while GrIS started to regain net mass from 121 ka[27].

### AIS and GrIS distinction.
Although GrIS did not affect LIG sea-level change significantly before 126.5–127 ka (Fig. 2a), the Red Sea and coral data compiled here imply that sea level crossed 0 m at 130–129.5 ka, during a rapid rise to a first highstand apex that was reached at ~127 (Fig. 2b, c, e, f). The Seychelles record indicates specifically that sea level reached 5.9 ± 1.7 m by 128.6 ± 0.8 ka[3]. We infer that both the first LIG rise above 0 m and the subsequent rapid rise between 129.5 and 127 ka resulted from AIS reduction. Similar qualitative inferences about an early-LIG AIS highstand contribution have been made previously[3,9,19],

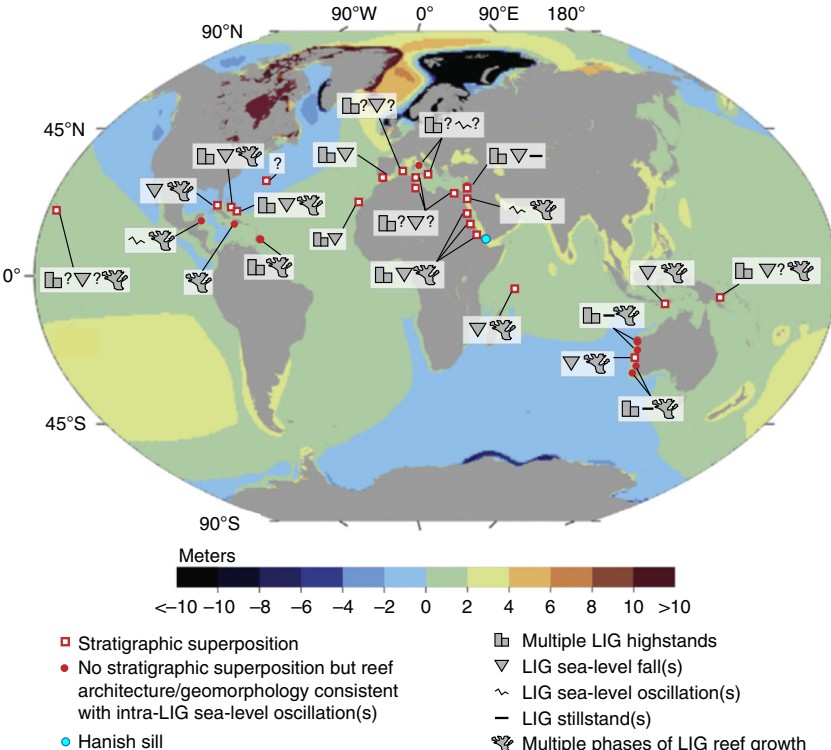

**Fig. 1** Global summary of stratigraphic evidence for Last Interglacial sea-level instability in coral-reef deposits and coastal-sediment sequences. Blue dot is the location of Hanish Sill, the constraining point for the Red Sea sea-level record. Red squares with white centres are stratigraphically superimposed coral reef or sedimentary archives for sea-level oscillations within the Last Interglacial (LIG). Solid red dots are locations where sea-level oscillations are inferred but where there is no stratigraphic superposition. The underlying map is of the difference between maximum Last Interglacial (LIG) relative sea level (RSL) values for glacio-isostatic adjustment (GIA) modelling results based on two contrasting ice models (ICE-1 and ICE-3) for the penultimate glaciation using Earth model E1 (VM1-like set up). The ICE-1 model is a version of the ICE-5G ice history (LGM-like), whereas ICE-3 has both reduced total ice volume relative to ICE-1, and a different ice-mass distribution (i.e., a smaller North American Ice Sheet complex and larger Eurasian Ice Sheet) that is consistent with glaciological reconstructions of the penultimate glacial period[4]

including attribution to sustained heat advection to Antarctica during Heinrich Stadial 11 (HS11; 135–130 ka)[19], when a northern hemisphere deglaciation pulse (~70 m sea-level rise in 5000 years) caused overturning-circulation shutdown[28], a widespread North Atlantic cold event, and southern hemisphere warming (Fig. 2d). Here we present a quantitative AIS and GrIS separation with comprehensively evaluated uncertainties.

First, we determine centennial-scale LIG sea-level variability from the continuous (and contiguous) single-core RSL record of central Red Sea core KL11 on our new Red Sea LIG age model. We validate this record with new data for high-accumulation-rate core KL23 from the northern Red Sea; i.e., from a physically separate setting than KL11 (Methods) (Fig. 2e). Given this validation, we continue with KL11 alone because it remains the most detailed record from the best-constrained (central) location in the Red Sea RSL quantification method, where $\delta^{18}O$ is least affected by either Gulf of Aden inflow effects in the south, or northern Red Sea convective overturning and Mediterranean-derived weather systems in the north[16,29].

Second, we perform a Monte Carlo (MC)-style probabilistic analysis of the KL11 record (Fig. 2f), which accounts for all uncertainties in individual-sample RSL and age estimates (cf. blue cross in Fig. 2e). This procedure mimics that applied previously to the Red Sea stack[10,18], but now contains an additional criterion of strict stratigraphic coherence (Methods). The analysis leads to statistical uncertainty reduction based on datapoint characteristics, density, and stratigraphy. Remaining RSL uncertainties are ±2.0 to 2.5 m for the 95% probability zone of the probability maximum (PM, modal value; Fig. 2f; Methods).

Both PM and median reveal an initial RSL rise from ~129.5 to ~127 ka to a highstand apex centred on ~127 ka, followed by a drop to a lowstand centred on 125–124 ka at a few metres below 0 m, and then a small return to a minor peak above 0 m at ~123 ka (Fig. 2f). To quantify AIS contributions, we apply a first-order glacio-isostatic correction (with uncertainties) to translate the record from RSL to global mean sea level (GMSL) (Supplementary Note 4) (Fig. 3a), and then subtract the GrIS-contribution records (Figs. 2a and 3b). Our results quantify significant asynchrony and amplitude-differences between GrIS and AIS ice-volume changes during the LIG (Fig. 3b, c). A caveat applies in intervals where the reconstructed AIS sea-level record drops below −10 m, because at that stage the maximum AIS growth limit is approximated (AIS growth is limited by Antarctic continental shelf edges). Whenever the reconstructed AIS sea-level record falls below −10 m (notably after ~119 ka), North American and/or Eurasian ice-sheet growth contributions likely became important. This timing agrees with a surface-ocean change south of Iceland from warm to colder conditions[27].

**Intra-LIG sea-level variability**. Red Sea intra-LIG variations are generally consistent (within uncertainties) in timing with apparent sea-level variations in the well-dated and stratigraphically coherent coral data from the Seychelles, and Bahamas[3,22,23], but with larger amplitudes. Northwestern Red Sea reef and coastal-sequence architecture reconstructions offer both timing and amplitude agreement (although age control needs refining)[30,31] (Supplementary Note 1). The reef-architecture study in particular[30] indicates an early-LIG sea-level rise with a post-128-ka

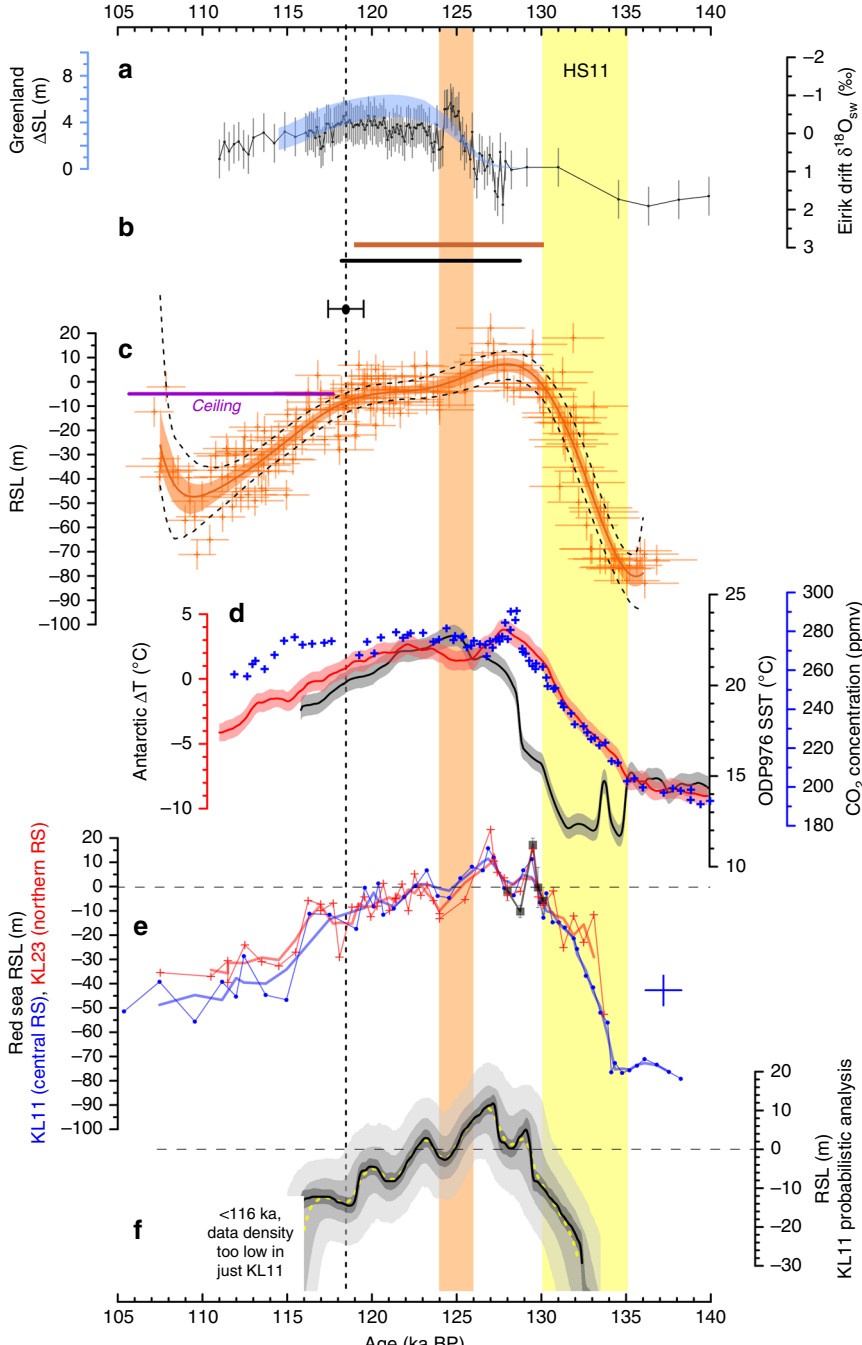

**Fig. 2** Variability in Last Interglacial sea-level time-series. Yellow bar: time-interval of Heinrich Stadial 11 (HS11)[19]. Orange bar: approximate interval of temporary sea-level drop in various records. Dashed line: end of main LIG highstand set to 118.5 ka (cross-bar indicates 95% confidence limits of ±1.2 ka), based on compilations in **b** and the speleothem sea-level "ceiling" (**c**). **a** GrIS contributions to sea level from a model-based assessment of Greenland ice-core data (blue)[9], and changes in surface sea-water δ18O at Eirik Drift (black; this study) with uncertainties (2σ) determined from underpinning δ18O and Mg/Ca measurement uncertainties and Mg/Ca calibration uncertainties. **b** Ninety-five per cent probability interval for coral sea-level markers above 0 m[11] (brown), and LIG duration from a previous compilation (black)[26]. **c** Red Sea RSL stack (red, including KL23) with 1σ error bars. Smoothings are shown to highlight general trends only, and represent simple polynomial regressions with 68% and 95% confidence limits (orange shading and black dashes, respectively). Purple line indicates the sea-level "ceiling" indicated by subaerial speleothem growth (Yucatan)[24]. **d** Probability maximum (PM, lines) and its 95% confidence interval for Antarctic temperature changes (red)[68], and proxy for eastern Atlantic water temperature (ODP976, grey)[69]. Blue crosses: composite record of atmospheric CO2 concentrations from Antarctic ice cores[19]. **e** Individual records for Red Sea cores KL11 (blue, dots) and KL23 (red, plusses), with 300-year moving Gaussian smoothings (as used in ref. [1]). Also shown is a replication exercise to validate the single-sample earliest-LIG peak in KL23 (grey, filled squares) with 1 standard error intervals (bars, σ/√{N}, based on N = 5, 5, 4, 4, and 5 replications, from youngest to oldest sample, respectively). Separate blue cross indicates typical uncertainties (1σ) in individual KL11 datapoints prior to probabilistic analysis of the record. **f** Probabilistic analysis of the KL11 Red Sea RSL record, taking into account the strict stratigraphic coherence of this record. Results are reported for the median (50th percentile, dashed yellow), PM (modal value, black), the 95% probability interval of the PM (dark grey shading), and both the 68% and 95% probability intervals for individual datapoints (intermediate and light grey shading, respectively)

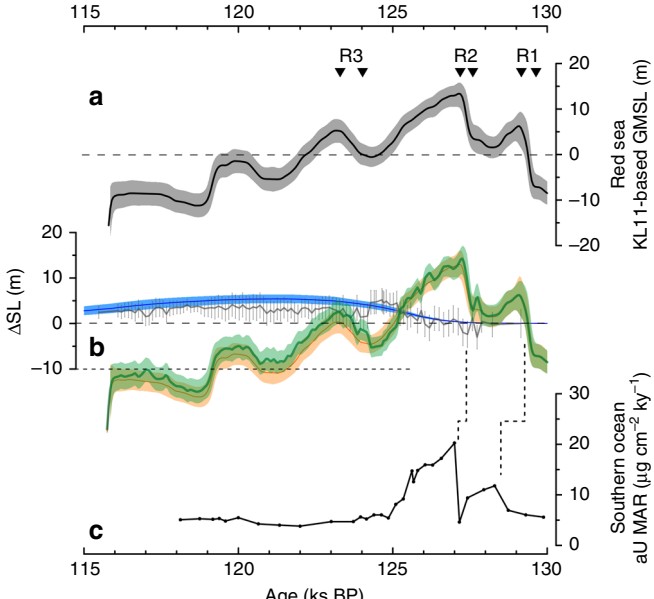

**Fig. 3** Identification of Greenland Ice Sheet and Antarctic Ice Sheet contributions to Last Interglacial sea-level variations. **a** Global Mean Sea Level (GMSL) approximation based on the probabilistically assessed KL11 PM (black line) and its 95% probability interval (grey). This record is shown in terms of RSL in Fig. 2f, but here includes the glacio-isostatic correction and its propagated uncertainty. Black triangles identify limits between which sea-level rises R1, R2, and R3 were measured. Rates of rise with 95% bounds: R1 = 2.8 (1.2–3.7) m c⁻¹; R2 = 2.3 (0.9–3.5) m c⁻¹; R3 = 0.6 (0.1–1.3) m c⁻¹. **b** Blue: GrIS sea-level contribution from the model-data assimilation of ref. 9 (shading represents the 95% probability interval). Grey: GrIS contribution based on Eirik Drift $\delta^{18}O_{sw}$. Uncertainties as in Fig. 2a. Orange: AIS contribution from subtraction of the blue GrIS reconstruction from the record in **a**. Green: AIS contribution found by subtracting the grey GrIS reconstruction from the record in **a**. Orange and green AIS reconstructions are shown as medians (lines) and 95% confidence intervals (shading). Reconstructed AIS contributions cross downward through a fine dashed when they fall below –10 m, which indicates a rough maximum AIS growth limit in terms of sea-level lowering (AIS growth is limited by Antarctic continental shelf edges). When the green/orange curves fall below these limits, North American and/or Eurasian ice-sheet growth is likely implied. The key result from the present study lies in identification of GrIS and AIS sea-level contributions above 0 m. **c** Southern Ocean ODP (Ocean Drilling Program) Site 1094 authigenic uranium mass accumulation rates, on its original, Antarctic Ice Core Chronology (AICC2012) tuned, age model. Dashed lines indicate potential offsets (within uncertainties) between the ODP 1094 AICC2012-based chronology36 and our LIG chronology (see refs. 10,19 and this study)

culmination at 5–10 m above present, followed by a millennial-scale ~10 m sea-level drop to a lowstand centred on ~124 ka.

In more detail, the probabilistic Red Sea record suggests a statistically robust dual substructure within the initial LIG sea-level rise (Fig. 2f), which is replicated between Red Sea records (Fig. 2e). It is not (yet) supported in wider global evidence (Methods, Supplementary Note 1), but there are indications that certain systems may have recorded it independently. For example, southwestern Red Sea reef-architecture reveals two main reef phases with a superimposed minor patch-reef phase1,32, reaching total thicknesses up to 10 m. But more precise dating and support from other locations are needed to be conclusive. In this context, we calculate with a basic fringing-reef accretion model that the rapid rises and short highstands inferred here (Fig. 2e, f) may have left limited expressions in reef systems, except for rare ones

with exceptionally high accretion rates, or where rapid crustal uplift offset some of the rapid sea-level rises (Supplementary Note 5). Hence, we consider wider palaeoceanographic evidence to evaluate the suggested sea-level history.

**Palaeoceanographic support**. AIS meltwater pulses implied by sea-level rises R1 and R2 (Fig. 2f) should have left detectable signals around Antarctica. The early-LIG AIS sea-level contribution occurred immediately after Heinrich Stadial (HS) 11, when overturning circulation had recovered from a collapsed HS11 state (Figs. 2–4)28. This likely enhanced advection of relatively warm northern-sourced deep water into the Circumpolar Deep Water (CDW), which impinges on the AIS. At the same time, there was a peak in Antarctic surface temperatures (Figs. 2d and 4c) and Southern Ocean sea surface temperatures (ODP Site 1094 TEX₈₆L, ODP Site 1089 planktic foraminiferal $\delta^{18}O$) (Fig. 4c–e), and Southern Ocean sea ice was reduced (Fig. 4b). We infer that early-LIG AIS retreat resulted from both atmospheric and (subsurface) oceanic warming, which—together with minimal sea ice (important for shielding Antarctic ice shelves from warm circumpolar waters, e.g., ref. 33)—drove enhanced subglacial melting rates and ice-shelf destabilisation, and thus strong AIS sea-level contributions between 130 and 125 ka.

Wider palaeoceanographic evidence can be used to test the concept that major AIS melt will provide freshwater to the ocean surface, which density-stratifies the near-continental Southern ocean, impeding Antarctic Bottom Water (AABW) formation34,35, which in turn will lead to reduced AABW ventilation/oxygenation and an increase in North Atlantic Deep Water (NADW) proportion vs. AABW proportion in the Atlantic Ocean28,36. Thus, we infer strong support for early-LIG AIS melt from palaeoceanographic observations. For example, an anomaly in authigenic uranium mass-accumulation rates (aU MAR) in Southern Ocean ODP Site 1094 has been attributed to bottom-water deoxygenation (AABW reduction/stagnation), due to strong Antarctic meltwater releases and consequent water-column stratification36 (Figs. 3c and 4g). Also, increased bottom-water $\delta^{13}C$, due to expansion of high-$\delta^{13}C$ NADW at the expense of low-$\delta^{13}C$ AABW, occurred at the end of HS11 in both the abyssal North Atlantic (ODP Site 1063, core MD03-2664) and South Atlantic (Sites 1089 and 1094) (Fig. 4i). Moreover, $\epsilon_{Nd}$ changes in Site 1063 (ref. 28) support the $\delta^{13}C$ interpretation (Fig. 4h). Given that intensification of relatively warm NADW likely plays a key role in subglacial melting and resultant AABW source-water freshening33,37, we infer a positive feedback. In this feedback, meltwater-induced AABW reduction warmed CDW through increased admixture of relatively warm NADW, which then caused further subglacial melting and AABW source-water freshening, driving additional AABW decline. Finally, a distinct early-LIG minimum in the Site 1089 planktic–benthic foraminiferal $\delta^{18}O$ gradient indicates a persistent surface buoyancy anomaly, which agrees with strong AIS meltwater input38 (Fig. 4c–f). Surface buoyancy/stratification increase would restrict air–sea exchange and subsurface heat loss. Analogous to explanations offered for high melt rates in some regions of Antarctica today and for even higher melt rates in a warmer future climate39, we therefore propose another positive feedback for the LIG, in which melt-stratification led to subsurface ocean warming, which then intensified ice-shelf melting.

Finally, we note that the aU MAR variations in Southern Ocean Site 1094 (ref. 36) also agree in more detail with our inferred dual substructure in the AIS-related early-LIG highstand (Fig. 3b, c). It is not yet possible to eliminate robustly the inferred offsets (which fall within uncertainties) between the ODP 1094

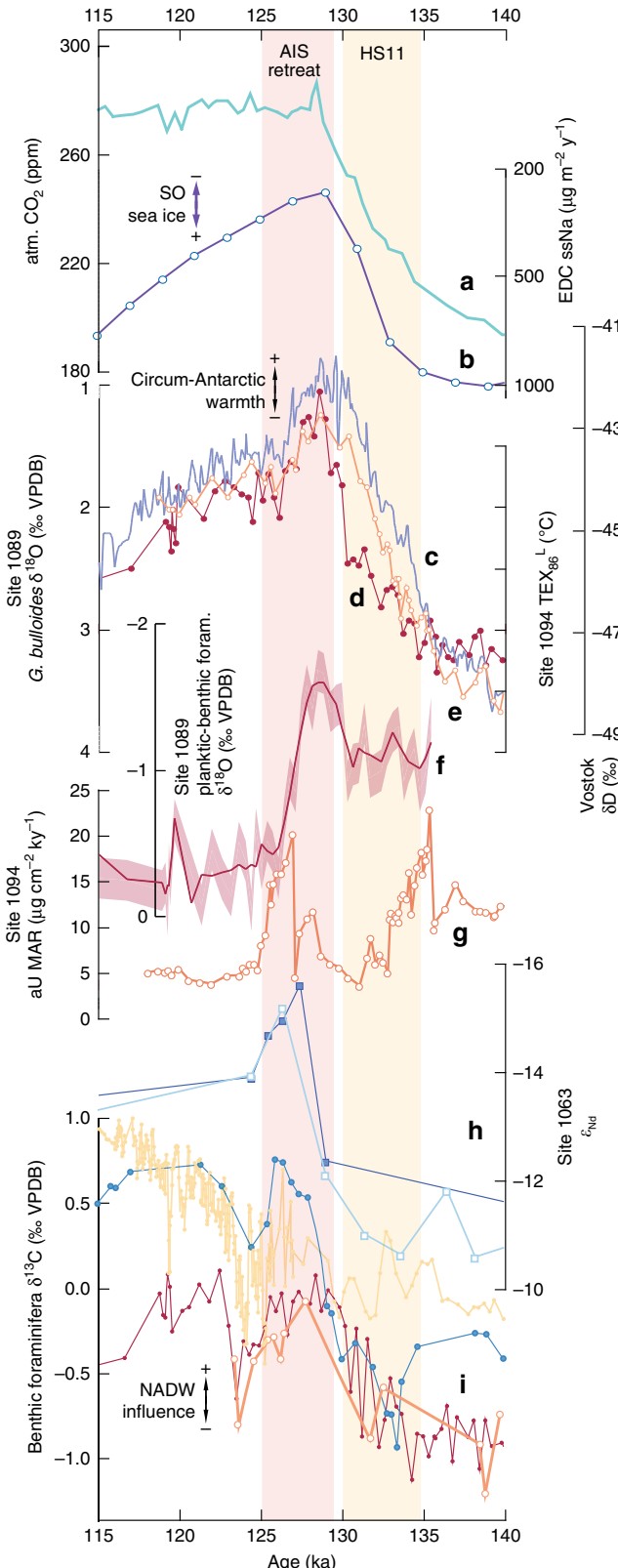

AICC2012-based chronology[36] and our LIG chronology (see refs. [10,19] and this study) (Fig. 3b, c), but the offsets may also (partly) arise from time-lags between meltwater input at the surface and oxygenation decline at the sea floor. Given the position of ODP Site 1094 (South Atlantic sector), the aU MAR record may be to some extent site-specific, in which case it

suggests a likely meltwater source from the West Antarctic Ice Sheet (WAIS). The lack of later aU MAR spikes for our further inferred AIS contribution may then suggest either that most of WAIS had been lost during the earliest LIG, or that it had at least retreated far enough to stop contributions as is also indicated by ice-sheet studies[14,40–43].

**Fig. 4** Timing of Antarctic Ice Sheet retreat relative to circum-Antarctic climate and ocean warming. LIG records of **a**. Antarctic ice core composite atmospheric $CO_2$ (ref. [70]), **b** EPICA Dome C sea-salt Na flux (on a logarithmic scale), which reflects Southern Ocean sea-ice extent[71], **c** Vostok δD (lilac)[67,72], **d** Site 1089 planktic foraminiferal (*G. bulloides*) δ18O (red)[38], **e** Site 1094 TEX86 L-based sea surface temperatures (orange)[36], **f** Site 1089 planktic minus benthic foraminiferal δ18O (‰) plotted as 3-point running mean (red) and sample average including combined 1-sigma uncertainty (light red shading)[38], **g** Site 1094 authigenic uranium (aU) accumulation where higher values indicate bottom-water deoxygenation[36], **h** Site 1063 $ε_{Nd}$ (dark blue, measured by MC-ICP-MS; light blue, measured by TIMS)[28], and **i** bottom-water δ13C records from Site 1063 (blue, 3-point running mean, based on benthic foraminifera *Cibicidoides wuellerstorfi*, *Melonis pompilioides*, and *Oridorsalis*)[28], MD03–2664 (yellow, 3-point running mean, *C. wuellerstorfi*)[73], Site 1089 (red, *C. wuellerstorfi*)[36], and Site 1094 (orange, *C. wuellerstorfi*)[36]. **h** and **i** Indicate North Atlantic Deep Water (NADW) influence as denoted. Map inset includes marine core locations, plotted using Ocean Data View (https://odv.awi.de)

## Discussion

The summarised suite of palaeoceanographic observations offers strong support to our reconstruction that early-LIG sea-level rise above 0 m derived from the AIS, and that this meltwater input occurred in several distinct pulses. Interruption of the rapid AIS mass-loss rate during the main phase of ice-sheet/shelf reduction may reflect negative feedbacks of isostatic rebound and resultant ice-shelf re-grounding that temporarily limited ice-mass loss (e.g., refs. [44–49]). The sea-level-lowering rates we find in between the LIG rapid-rise events range between multi-centennial means of −0.23 and −0.63 m c$^{-1}$ (with peaks up to −1 m c$^{-1}$) (Fig. 2g, Supplementary Fig. 10). These imply high rates of global net ice-volume growth, but we note that LIG accumulation rates over the AIS may have been ~30% higher than present[50] (Supplementary Note 6).

Our record (Fig. 3a) indicates a first sea-level rise (R1) above 0 m at event-mean values of 2.8 (1.2–3.7) m c$^{-1}$, followed by R2 at 2.3 (0.9–3.5) m c$^{-1}$, and R3 at 0.6 (0.1–1.3) m c$^{-1}$, where the ranges in brackets reflect the 95% probability bounds. These values lend credibility to similar rates inferred from ice modelling that includes certain ice-shelf hydrofracturing and ice-cliff collapse paramerisations[51]. These processes remain debated, but the apparent reality of such extreme rates in pre-anthropogenic times —when climate forcing was slower, weaker, and more hemispherically asynchronous than today—increases the likelihood that such poorly understood mechanisms may be activated under anthropogenic global warming, to yield extreme sea-level rise.

In conclusion, we have reconstructed (Fig. 3) an initial sea-level highstand (above 0 m) at ~129.5 to ~124.5 ka, which derived almost exclusively from the AIS (in agreement with palaeoceanographic evidence), and which reached its highstand apex at around 127 ka. We find that the rise toward the apex occurred in two distinct phases, which also agrees with a palaeoceanographic record of AABW ventilation changes. Following the apex at ~127 ka, we reconstruct a sea-level drop to a relative lowstand centred on 125–124 ka, which in turn gave way to a minor rise toward a small peak at or just above 0 m at ~123 ka. GrIS contributions were differently distributed through time. These contributions slowly ramped up from ~127 ka onward, reaching maximum, sustained contributions to LIG sea level from ~124 ka until the end of the LIG. Thus, we quantitatively reconstruct that there was strong asynchrony in the AIS and GrIS contributions to the LIG highstand, with an AIS-derived maximum that spanned from ~129.5 to ~124.5 ka, a low centred on 125–124 ka, and variable, joint AIS + GrIS influences from ~124 to ~119 ka.

We observe rapid rates of sea-level change within the LIG. These may reflect complex interactions through time between: (a) enhanced accumulation during a regionally warmer-than-present interglacial[50]; (b) persistent dynamic ice-loss due to long-term heat accumulation (e.g., ref. [19]); (c) negative glacio-isostatic feedbacks to ice-mass loss (e.g., refs. [44–49]); and (d) positive oceanic feedbacks to Antarctic meltwater releases (Discussion, and refs. [35,52]). Similar sequences may develop in future, given that warmer CDW is encroaching onto Antarctic shelves, so that

future sea-level rise may become driven by increasingly rapid mass-loss from the extant AIS ice sheet[53–56], in addition to the well-observed GrIS contribution[57,58].

Finally, we infer intra-LIG sea-level rises with event-mean rates of rise of 2.8, 2.3, and 0.6 m c$^{-1}$. Such high pre-anthropogenic values lend credibility to similar rates inferred from some ice-modelling approaches[51]. The apparent reality of such extreme pre-anthropogenic rates increases the likelihood of extreme sea-level rise in future centuries.

## Methods

**Red Sea relative sea level record.** The Red Sea RSL record derives from contiguous sampling of sediment cores and, thus, has tighter stratigraphic control than samplings of reef systems, which consist of more complex three-dimensional frameworks. Red Sea sediment cores consist of beige to dark brown hemipelagic mud and silt, with high wind-blown dust contents in glacial/cold intervals and lower wind-blown dust contents in interglacial intervals. This results in colour and sediment-geochemistry variations that allow straightforward assessment of bioturbation. This was found to be very limited in the cores used here, which agrees with extremely low numbers of benthic microfossils (benthic numbers per gram are an order of magnitude, or more, lower than planktonic numbers per gram[59], reaching two orders of magnitude lower in the LIG[60]), which in turn agree with extremely low Total Organic Carbon contents (at or below detection limit)[60]. With limited bioturbation, the stratigraphic coherence of the sediment record is well preserved.

The new KL23 δ18O analyses were performed on 30 specimens per sample of the planktonic foraminifer *Globigerinoides ruber* (white) from the 320 to 350 μm size fraction. Sample spacing and KL11-equivalent age model are indicated in the data file. Prior to analysis, foraminiferal tests were crushed and cleaned by brief ultrasonication in methanol. Measurements were performed at the Australian National University using a Thermo Scientific DELTA V Isotope Ratio Mass Spectrometer coupled with a KIEL IV Carbonate Device. Results are reported in per mil deviations from Vienna PeeDee Belemnite using NBS-19 and NBS-18 carbonate standards. External reproducibility (1σ) was always better than 0.08‰.

Red Sea carbonate δ18O is calculated into RSL variations using a polynomial fit to the method's mathematical solution[16,29] (see Supplement of ref. [17]). The Red Sea stack of records[17] was dated in detail through the last glacial cycle based on the U/Th dated Soreq Cave speleothem record[10]. Through the LIG, however, it was constrained only by interpolation between tie-points at 135 and 110 ka. The age model for the LIG-onset was later validated[19], yet the LIG-end remained to be better constrained. Here we make an important adjustment for the LIG-end, based on radiometrically dated criteria described in the main text. This assignment is based on a first-order assessment of the entire Red Sea stack using a simple polynomial and its 95% uncertainty envelope, and it is validated by the fact that in the more precise probabilistic analysis of KL11 alone, the 95% probability zone for individual datapoints (lightest grey) also crosses 0 m at 118.5 ka. We only use the latter in validation, to avoid circularity in the age-model assignment. This reassigns the level originally dated (by interpolation) at 123 ka in the Red Sea stack[10], to 118.5 ka with 95% uncertainty bounds of ±1.2, where the uncertainties relate to those of the original age model[10] (Fig. 2, Supplementary Fig. 2). Initial age uncertainties (at 95%) all derive from that study. Next, age interpolations using the adjusted chronological control point are performed probabilistically using a Monte-Carlo (MC)-style ($n = 2000$) sequence of Hermite splines that impose monotonic succession to avoid introduction of spurious age reversals (Supplementary Fig. 2). Our new chronology for the Red Sea LIG record implies low sediment accumulation rates without major fluctuations within the LIG (Supplementary Fig. 2). Finally, when performing the sea-level probabilistic assessment for core KL11, we use the newly diagnosed age uncertainties from Supplementary Fig. 2, which are wider (more conservative) through the interval 120–110 ka than the originals (Supplementary Fig. 2).

The two separate high-resolution LIG sea-level records from the Red Sea discussed here are an existing one from central Red Sea core KL11 (18°44.5′N, 39° 20.6′E)[1], and a new one from northern Red Sea core KL23 (25°44.9′N, 35°03.3′E). The new KL23 LIG record validates the KL11 record, but its early-LIG peak

comprises only one sample/datapoint. The validity of this peak was confirmed with a multiple replication exercise (Fig. 2e, grey).

Through its continuity, stratigraphic constraints, and consistently high signal-to-noise ratio and sea-level variations are identified in the Red Sea record with limited impacts from other factors[10,16–18,29]. However, the Red Sea sea-level record still is only a RSL record for the Hanish Sill, Bab-el-Mandab, and correction for glacio-isostatic influences is needed to obtain estimates of GMSL from this record (Supplementary Note 4). Following these corrections, we estimate AIS sea-level contributions by determining the difference between GMSL and two different estimates for the GrIS contribution (see ref. [9] and our Eirik Drift $\delta^{18}O_{sw}$ approach), with full propagation of the uncertainties involved (see below, and Supplementary Note 3).

The probabilistic analysis of the Red Sea core KL11 record (Fig. 2f) follows the same approach as for the Red Sea RSL stack[10,18], which gives similar results to an independent Bayesian approach using the same dataset[61]. The method uses the full probability distribution envelopes for both age and sea-level directions, as characterised by the mean and standard deviation per sample point (see blue cross in Fig. 2e for these $1\sigma$ limits in KL11), and performs 5000 MC-style resamplings of the record. During this resampling, we here apply an additional criterion of strict stratigraphic coherence within the contiguously sampled KL11 record (allowing no age reversals during MC-resampling). The resultant suite of MC simulations is then analysed at set time-steps to identify the probability maximum (modal value, with 95% probability window that depends on how well-defined the modal value is), median, and the 16th, 84th, 2.5th, and 97.5th percentiles that demarcate the 68% and 95% probability zones of the total MC-resampled distribution of individual-sample points (Fig. 2f). Because of the stratigraphic coherence in the KL11 record considered here, the modal value (and median) in each time-step probability distribution through the MC simulations is tightly constrained, with the mode (probability maximum) typically defined within 95% bounds of only ±2 to 2.5 m. In the earlier studies for the Red Sea stack[10,18], this was ±6 m, because a stack of different records does not preserve strict stratigraphic coherence from one datapoint to the next, so that relative age uncertainties between datapoints remained much larger than in our new record.

**Eirik Drift surface sea-water $\delta^{18}O$ record ($\delta^{18}O_{sw}$).** Our Eirik Drift surface sea-water $\delta^{18}O$ record ($\delta^{18}O_{sw}$) was determined for core MD03-2664 (57°26′N, 48°36′W, 3442 m) using the palaeotemperature equation of ref. [62], with a Vienna PeeDee Belemnite to Standard Mean Ocean Water standards conversion of 0.27‰, using $\delta^{18}O$ (ref. [63]) and Mg/Ca temperature data[64] for the planktonic foraminiferal species *Neogloboquadrina pachyderma* (sinistral; 150–250 µm size fraction), on the chronology of ref. [64]. Previously published estimates for $\delta^{18}O_{sw}$ covered only late MIS 6 and early MIS 5e (2600–2850 cm core depth[63]), and are supplemented here with new estimates for core depths ranging between 2350 and 2600 cm. Even today, the location of MD03-2664 is dominated by currents carrying admixtures of $^{16}O$-enriched Greenland melt water, with increased melt admixtures causing more negative $\delta^{18}O_{sw}$ values[65,66]. Specifically, $\delta^{18}O_{sw}$ at this site is highly sensitive to changes in the net freshwater $\delta^{18}O$ endmember[65]. Less GrIS meltwater discharge and relative dominance of sea-ice meltwater yield a less negative net freshwater endmember $\delta^{18}O$, whereas the opposite yields a very negative net freshwater endmember $\delta^{18}O$ (see ref. [65] and references therein). Regional freshwater end-member changes span a range of ~10‰ or more, so while marine endmember changes are <0.5‰[65], sustained MD03-2664 $\delta^{18}O_{sw}$ changes reflect net freshwater component changes, and therefore mainly GrIS melt. Using an endmember mixing model, and fully propagating generous uncertainties, we find that (all else being constant) the observed −1.3‰ $\delta^{18}O_{sw}$ change in MD03-2664 corresponds to 4 ± 1.2 m GrIS-derived sea-level rise (Supplementary Note 3).

## Data availability

The new Red Sea KL23 $\delta^{18}O$ and sea level data, Eirik Drift $\delta^{18}O_{sw}$ data supporting the findings of this study, and source data for Figs. 2 and 3, are provided with the paper as a Source Data file [https://doi.org/10.6084/m9.figshare.9790844] and via http://www.highstand.org. Further information is available from the corresponding author upon reasonable request.

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

## Acknowledgements

This research contributes to Australian Research Council Laureate Fellowship FL120100050 (to E.J.R.). UiB contribution (to E.V.G., N.I., K.K. and U.N.) supported by RCN project THRESHOLDS (25496). G.M. acknowledges generous support from the University of Vigo. All plotted new data will be made openly available via http://www.highstand.org/erohling/ejrhome.htm.

## Author contributions

E.J.R. and F.D.H. led the research. K.M.G., G.M., F.W. and J.Y. added wider documentation and context. H.S. contributed core curation, sampling, and processing assistance. E.V.G., N.I., K.K., U.N. and Y.R. provided new oxygen isotope and microfossil shell chemistry records for Eirik Drift. A.P.R. helped shape the initial concept and focussed the presentation. All co-authors assisted in producing the manuscript.

## Competing interests

The authors declare no competing interests.
