## [Peer Review File · Nature Communications]

Reviewers' comments:

Reviewer #1 (Remarks to the Author):

General

The paper makes a great attempt to put most of the observational based evidence for LIG sea-level variability together in one publication. For this, the authors are generously providing a wealth of details in the SI that make it possible to follow the argumentation.

The paper explicitly claims novelty in two aspects: 1) first quantitative, records-based, separation of GrIS and AIS contributions and 2) first time quantify significant asynchrony and amplitude-differences between GrIS and AIS ice-volume changes during the LIG. While I judge the actual novelty in both aspects less important than originally claimed (see specific comments L137 and L141 below and Kopp et al., 2009 and 2013), I do believe the paper to be of interest to the community and the wider field.

In my understanding, the paper could be improved by better representing missing links in our understanding of LIG sea-level. Without modification, the paper could be read to suggest that coral and sediment records are finally reconciled and in line with GrIS and AIS modelling evidence, which is clearly not the case. This is most obvious to me for the rates of ice mass build-up, which cannot be reconciled with current understanding of ice sheet physics (see comment L192), but is also the case for the mass loss rates and large SL variability in general.

With some improvements already mentioned and as detailed further below in my specific comments, I believe the manuscript would be an interesting contribution to the discussion on LIG sea-level.

Specific comments

L23 A possible contribution of GIC and thermal expansion is probably small compared to the uncertainties discussed here, but should nevertheless be mentioned somewhere in the text as a basis for the evoked residual argument. More importantly, much larger contributions may not be fully excluded from other NH ice sheets than the GrIS after 130 kyr and before 115 kyr BP (see e.g. Kopp et al., 2009).

L90 I fully agree with the interpretation of a late Greenland contribution. Additional support for that may be drawn from the reconstructed temperature evolution of the NEEM ice core record (Dahl-Jensen et al., 2013). To first order, one would expect a GrIS melting (and increasing SL contribution) as long as the temperature anomaly is above zero (until ~ 118 kyr). Aside from that, I find the similarity between the presented Eirik Drift reconstruction and the Yau results rather poor. Although uncertainty bands overlap in most cases, the central estimates differ by a factor 2 (by eye ~ 2 m) in large parts of the LIG period, which is a big difference for a Greenland reconstruction. I suggest to change the wording from adding support "for this GrIS reconstruction" to adding support "for a late Greenland contribution".

L111 I think it would help to clarify that the separation of GrIS and AIS signals is a simple difference of two records (with GrIS much smaller than the global and uncertainty on the latter of similar magnitude as the GrIS) and clarify the underlying assumptions: 1) the (corrected) Red Sea record has to be a good predictor for global SL and 2) that $GL-GrIS = AIS$. Note again contributions from glaciers and ice caps and other NH ice sheets.

L129 Why are the RSL uncertainty ranges of ± 2.0 to 2.5 so small compared to the much larger sample based ranges? This is called "statistical uncertainty reduction" in the text. I believe it would be needed to elaborate further on this point, because most of the conclusions from this work crucially depend on it. Large SL variability, large derived mass loss rates and large derived mass gain rates are all highly disputed and problematic to reconcile with other evidence. In other words, with larger uncertainty bands, it would be possible to find a SL curve that avoids most of the

controversial aspects of the record. I believe it should be also in the interest of the authors to make it as clear as possible why and how sure we can be about these narrow uncertainty bands. Conversely and as mentioned earlier, 2-2.5 m uncertainty is already quite large compared to contributions that may be expected from Greenland, which limits the usefulness of the residual calculations needed to derive the Antarctic signal.

L137 As a note on "Our results for the first time quantify significant asynchrony and amplitude-differences between GrIS and AIS ice-volume changes during the LIG":

I believe the results in Kopp et al. (2009) that distinguish NH and SH contributions in a probabilistic framework shows both asynchrony and amplitude-difference. The wording may need to be modified to reflect that.

L141 Intra-LIG variations and the importance of different reconstructions (e.g. Red Sea vs coral) has also been discussed in Kopp et al. (2013). It seems appropriate to evaluate the findings in resonance with that work.

L171 If the uranium mass-accumulation-rate signal in the SO can be attributed to Antarctic meltwater releases, why do the later events during the LIG do not show at all in the record in Figure 3? I believe this requires some further discussion to justify the record as support for AIS mass fluctuations.

L181 A peak (meaning a relative maximum) in atmospheric CO₂ levels and Antarctic surface temperatures does not necessarily imply enough atmospheric forcing to lead to considerable mass loss by surface processes. The atmospheric CO₂ forcing during the LIG compared to today is very limited and the AIS in its present state is in most places far (>5 deg) from considerable surface melt. I suggest to reformulate to put an emphasis on the ocean forcing, possibly supported by atmospheric forcing as a secondary effect.

L189 It is not clear to me why and how the GrIS comes into the discussion here, which deals exclusively with AIS processes so far. I'd suggest to remove reference to GrIS.

L192 The back-of-the-envelope calculation for ice buildup is flawed and not in line with the reality of ice sheet physics. Both ice sheets always have a mass wastage term, which compensates the received accumulation. The Greenland ice sheet constantly loses mass either by surface melting at the margins (when small) and/or due to iceberg calving (especially when the ice sheet is big). The Antarctic ice sheet must be assumed as largely marine based during the LIG with considerable calving, which continues at any time, so accumulation cannot be regarded separately to explain the regrowth.

I would suggest to remove this part from the manuscript and acknowledge that there exists a real problem to reconcile the large sea-level variations in your records with current understanding of ice sheet physics.

L199 Analysis from Kopp et al. (2013) concludes that it is "unlikely that the rate exceeded 7 m kyr⁻¹ and extremely unlikely that it exceeded 11 m kyr⁻¹". There is clearly a difference if rates are evaluated on centennial or millennial time scales, but it may also be questioned if your record has the time resolution to justify centennial time scale analysis. At any rate, I think it would be needed to put your results in perspective of these findings.

While the mentioned (unphysical) ice-cliff collapse parameterisations have been included in models to produce larger LIG sea-level contributions, they remain highly disputed due to their ad-hoc nature. Furthermore, the large rates of sea-level rise in DeConto and Pollard (2016) can only be found in the initial deglacial transition into the LIG (see their figure 3a), while during the LIG (where you diagnose the largest rates from your record) changes happen much more gradually. It is not surprising that large rates can be found for a full glacial-interglacial transition, while it is much more difficult to conceive such large rates from an ice sheet that is already reduced in size and also sees a forcing (oceanic and atmospheric) with less contrast.

L301 If I understand the following correctly, the LIG record is still constrained by interpolation between the same amount of tie-points, even if they have been improved compared to earlier interpretations. If this is the case, this should be made clear.

Figures

There is an overlapping shading in panel b of Figure 3 which may not be intended. Otherwise, figures are clear to me.

References

Dahl-Jensen, D., Albert, M. R., Aldahan, A., Azuma, N., Balslev-Clausen, D., Baumgartner, M., . . . Zheng, J. (2013). Eemian interglacial reconstructed from a Greenland folded ice core. *Nature*, 493(7433), 489-494. doi:10.1038/nature11789

DeConto, R. M., & Pollard, D. (2016). Contribution of Antarctica to past and future sea-level rise. *Nature*, 531(7596), 591-597. doi:10.1038/nature17145

Kopp, R. E., Simons, F. J., Mitrovica, J. X., Maloof, A. C., & Oppenheimer, M. (2009). Probabilistic assessment of sea level during the last interglacial stage. *Nature*, 462(7275), 863-867. doi:10.1038/nature08686

Kopp, R. E., Simons, F. J., Mitrovica, J. X., Maloof, A. C., & Oppenheimer, M. (2013). A probabilistic assessment of sea level variations within the last interglacial stage. *Geophysical Journal International*, 193(2), 711-716. doi:10.1093/gji/ggt029

Reviewer #2 (Remarks to the Author):

This paper presents a SL model of the LIG from a Red Sea sediment core reconstruction with improved chronological constraints. It validates this SL model with studies of SL derived from selected LIG coral reef units (Seychelles and the Bahamas), claiming that they represent the 'best sites' for comparison. Once validated, it then goes on to highlight implications of the model for ice sheet contributions.

I will comment on this comparison between the LIG reefs and the Red Sea reconstruction (RSR), both because this validation is the main focus of the claims made, and because this falls within my field of expertise. I assume that others have assessed both the statistical modelling of sediment-core data and methods used to derive an accurate and precise SL reconstruction.

These authors claim that the RSR shows a SL maxima of +10 m between 130-126 ka, a rapid metre-scale fall between 125-124 ka, and then a second maxima between 123-118 ka. In other words a decaying double highstand. They claim this reconstruction is stratigraphically "stronger" than those derived from reefs yet fail to state uncertainties with sediment-cores records, such as theirs.

Next, they validate this decaying double highstand with reef data. They state that Seychelles reef data supports an early +10m highstand, yet offer no explanation as to why this site was chosen, other than it represents the "...best sites for combined stratigraphy and precise datings...". However, the age and stratigraphic data cited in Dutton et al study is not unambiguous and suffers from the same deficiencies as other studies. Specifically it fails to parameterise the interpretation of in-situ framework from clast deposits, fails to account for non-replication of intra-sample ages, and fails to consider stratigraphic consistency of ages and rates of framework development in both modern and Pleistocene reefs. Specifically, true age variation at individual sites ranges up to 4.5ka which is stratigraphically inconsistent and contrary to evidence from Holocene reef-accretion data showing coral ages in a 3 m vertical section varying by <1000 years (Edinger et al. 2007 EPSL 253, p37). The Vyverberg et al study is more rigorous but also fails to account for inconsistent stratigraphic ages. More importantly, it fails to consider the effect of the set-up from Southern Ocean swell which commonly produces coralgall development above mean SL (e.g. modern algal

ridges can develop up to 2 m above msl on Indo-Pacific reefs).

So the main reason these authors choose a comparison with the Seychelles is because it supposedly supports their claim of an early >6 m highstand. Their uncritical acceptance of these data, however, does them no favors and suggests confirmation bias. The same is true of the Bahamas studies cited: despite good data contesting their findings, these studies were chosen because they also claim a SL drop at a similar time to the RSR.

Next they address the problem that the highstand magnitude in the RSR is 2-4 m higher than in either the Seychelles or Bahamas. To account for this they cite work from the Egyptian Red Sea coast as supporting a higher LIG SL. Unfortunately they fail to state that LIG reef units along this coast vary in elevation between +8 to +18 m and thus have been influenced by neotectonic activity associated with the rifted Red-Sea margin (as is clearly stated in fig 4.4 of the paper they cite; see also Plaziat et al 1998, *In Sed & Tec in Rift Basins: Red Sea-Gulf of Aden* ISBN 0412 73490 7). This is why these works only state a general range of LIG SL between 5-10 m, and not because there is clear unequivocal evidence that SL was higher than other records.

So rather than presenting a balanced presentation of the state of LIG SL models and how they compare and contrast with the updated RSR, we are subject to only the parts that support their model. The authors might suggest that this is covered in the supplementary information, but the fact remains that the true uncertainty has been glossed over in the main text.

My suggestion to improve the manuscript would be to concentrate more on the uncertainty related to their RSR. For example, the 124ka reversal is not the only excursion shown by their data. Once these have been covered, it would be somewhat easier to introduce the various SL models based on the reef stratigraphy and have a balanced discussion about what parts of each are more or less uncertain. Unfortunately, given the still large uncertainties associated with SL records from the LIG, my feeling is that it's a little premature to start making important claims about which ice-sheet was to blame. From this perspective, I don't think the manuscript presents a substantial advance over what is already known.

Reviewer #3 (Remarks to the Author):

This work compares estimates of global mean sea level during the last interglacial from the Red Sea to inferred sea level contributions of the Greenland ice sheet to determine the sea level contribution of the Antarctic ice sheet over time. The Red Sea estimates build upon previous work by these authors, and new data is presented to address some of the standing controversy. The authors also present a new calculation for the Greenland ice sheet sea level contribution using published data from an ocean core near Greenland ice sheet. This new calculation agrees, within uncertainty, with estimates derived from ice cores on Greenland (Yau et al 2016). In both cases, Greenland is thought to contribute little to global mean sea level during the start of the last interglacial. However, the Red Sea estimates of global mean sea level (as well as many other less stratigraphically coherent records) suggest global mean sea level was highest during the start of the last interglacial. Therefore, there must be a significant contribution of Antarctic melt early during the early stages of the last interglacial. In fact, this specific conclusion is not new (ie. see conclusions in Yau et al 2016). However, since there are high frequency oscillations in the Red Sea record prior to significant melting of Greenland ice sheets, this work suggests that the high frequency changes in global mean sea level can be caused by a dynamics within a single ice sheet, and not just the interplay of more than one ice sheet. Ultimately, the Greenland contribution is very small compared to the size of the sea level signal recorded in the Red Sea data, so the Antarctic contribution ends up looking very similar to the Red Sea record. Therefore, a key component to this paper is the new data from core KL23, and the degree to which that core 'validates' the use of KL11 as a GMSL curve (which I will discuss in more detail below). As a whole, this work is a novel and significant contribution to our understanding of past ice sheet dynamics. The paper is very clearly written, the figures are effective, and the methods and supplementary material are detailed and extensive. While I have my own reserves about the magnitude of GMSL change suggested by the Red Sea data, the calculations presented here, and in previous work,

consider many of the important uncertainties (such as glacial isostatic adjustment), and I think that the community moves closer towards understanding the last interglacial and ice sheet dynamics by critically reading this paper.

Below are some more specific comments that may be helpful to the authors:

KL23 and KL11:

One of the controversies over KL11 in previous work was that another core in the Red Sea did not show the same high frequency change, and that the variability of change is similar in magnitude to the uncertainty. The new data presented here from the high accumulation rate core KL23 show the same high frequency change as KL11. This conclusion can be drawn from Figure 2e. However, I think this conclusion may be easier for the reader to see without the 300 year Gaussian smoothing lines.

Also, I am a little confused by the discrepancy between the (Grey) replicate tests and the red KL23 data. If I understand correctly, the Grey data represents multiple measurements from the same intervals of the KL23 core. The red data lies within the grey data uncertainty, except for the drop where the red data is much higher. Why is there such a difference here (10 meters of RSL)? I don't think this is a critical issue for the interpretations, but some clarity would be helpful. Maybe I missed something?

Lines 76-80 - Was GIA considered for the Yucatan cave deposits? If the same GIA models used for the Red Sea RSL curves are applied to the Yucatan data, does that significantly change the new age-model? I suspect it could for some of the GIA models, but it is difficult to test if those models are very useful at that location. I guess the background map in Figure 1 addresses this issue a little bit -- there is little difference in the RSL calculation between GIA models at the Yucatan location. The authors may decide to include a line pointing this out. I am not sure the difference would be so small if looking at time slices within the interglacial (not just the maximum RSL).

On lines 120-121, the authors state that the KL11 core is the most detailed and the 'best' core, so it will be the focus of the paper. I could guess why central (KL11) is better than the northern core, but a more explicit statement may help the reader.

Lines 137-139 - I suggest to delete the words 'for the first time' as the results in this paper stand on their own.

Line 348 - I think the authors meant Figure 2e.

-Blake Dyer

Response to reviewers

Rohling et al., “Asynchronous Antarctic and Greenland ice-volume contributions to the last interglacial sea-level highstand”

We thank the reviewers for their thorough and useful comments. Reviewer 1 commented that “*the paper makes a great attempt to put most of the observational based evidence for LIG sea-level variability together in one publication...generously providing a wealth of details in the SI that make it possible to follow the argumentation*” and that the manuscript is an “*interesting contribution to the discussion on LIG sea level*”. This was echoed Reviewer 3; “*the community moves closer towards understanding the last interglacial and ice sheet dynamics by reading this paper*” and states that the *Supplementary Information* is “*extensive and detailed*”. Reviewer 2 by own admission concentrated on the coral records, and may have overlooked the other evidence we present for Last Interglacial ice-sheet instability (possibly because it was in the *Supplement*, and we have therefore moved the palaeoceanographic support into the main paper), as well as the study’s stated focus on unravelling asynchronous contributions of the Antarctic and Greenland sheets. Reviewer 2 also wanted more on uncertainty, but Reviewer 3 stated that “*the calculations presented here, and in previous work, consider many of the important uncertainties.*” Still, we have added even stronger emphasis on where uncertainties come from (or are constrained), and how they propagate.

Our responses to specific comments are given below in blue, with actions highlighted in yellow.

REVIEWER 1

L23: contribution of glaciers, ice caps and thermostatic expansion

Regarding the thermosteric component, it is important to note that detailed temperature reconstruction exists only for the surface (especially where this concerns the time-evolution of temperature through the LIG). For the vast interior of the ocean, it does not exist. Noble gas estimates of mean ocean temperature are under way, but not published yet. Overall, however, the temperature difference relative the pre-industrial present were small. McKay *et al.*, 2011^{ref.1}, using modelling and palaeo data, imply minimal contribution of thermal expansion to LIG sea level and concluded that; “it seems unlikely that thermosteric sea-level rise exceeded 0.4 ± 0.3 m during the LIG”. Hoffman *et al.*, 2017^{ref.2} estimated a thermosteric contribution of 0.08 to 0.51 m to the LIG highstand using a coupled climate model. Hence, this component may have been negligible, up to perhaps half a metre or so, and this fall well within our uncertainty envelopes to our reconstructions, which are total sea-level values.

Ice caps and mountain glaciers: we are not aware of any conclusive LIG estimates of this in the literature. Yet, we expect this to be small (e.g., these processes contributed only 0.41 ± 0.08 mm/yr from 2003 to 2010, Jacob *et al.*, 2012^{ref.3}; and all modern ice-caps and glaciers contain only up to $\sim 0.6 \pm 0.1$ m sea level equivalent, Radi and Hock, 2010^{ref.4}, recent estimate suggests a sea-level contribution of 0.32 ± 0.08 from all extant mountain glaciers and ice caps, Farinotti *et al.*, 2019^{ref.5}). Thus, this again falls well within the uncertainties of our reconstructions.

We have added summary information about these components to the *Introduction*.

L90: Late Greenland contribution – similarity between our $\delta^{18}\text{O}_{\text{sw}}$ record and Yau *et al.*, 2016 GrIS reconstruction

Though the reviewer agrees with the general trend (in time) of the reconstructions we use, (s)he is concerned that the two approaches we use give central estimates that are somewhat different, even if the 95% probability envelopes overlap. We see this also, but given the cold facts of statistics, we do not feel that we are warranted to discuss differences in the central estimates exactly BECAUSE the 95% probability envelopes overlap. In our calculations of the LIG Antarctic contribution, we use BOTH of these estimates (see orange and green curves in *Figure 3b*). While we adhere to these principles of

statistical robustness, we really like the reviewer's suggestion of a change in wording, and have accordingly implemented that. We now say "...support for the inferred late Greenland contribution" where we originally had "...support for this GrIS reconstructions."

L111: Clarification of the separation of the GrIS and AIS signal

We have added a sentence to the *Methods* section to clarify this.

To specifically answer to the assumptions highlighted by the reviewer:

(a) "Red Sea is a good predictor of global sea level." This is dealt with in the *Supplementary Information/Methods* taking into account that Hanish Sill experiences some attenuation of relative sea level (RSL) changes with respect to global mean sea level (GMSL) changes, due to glacio-isostatic effects (GIA) (Milne and Mitrović, 2008^{ref.6}; Williams, 2016^{ref.7}). Global ice volume changes are by far the dominant control on the GIA response at Hanish Sill (Lambeck *et al.*, 2011^{ref.8}; Williams, 2016^{ref.7}; Rohling *et al.*, 2017^{ref.9}). We have fully taken the GIA processes into account when separating the GrIS and AIS signals (i.e., the Red Sea record is GIA corrected to GMSL first).

(b) "GMSL-GrIS = AIS"; To first order (i.e., within stated uncertainties), this is a reasonable assumption, given the small (unresolvable, see above) contribution from thermosteric effects and mountain glaciers/ice caps. This argument is given in the *Methods*.

L129: Red Sea statistical uncertainty reduction

In the *Methods* section, we explained how we analysed the probability intervals to the Red Sea sea-level record, namely following a slightly modified version of the approach by Grant *et al.* (2012^{ref.12}; described in their supplement). Those authors state, in their supplement: "To determine confidence limits to RSL that fully account for the combined uncertainties in both age and sea-level reconstruction (main-text Fig. 2), we have defined normal distributions around each datapoint, based on the mean datapoint values and their standard deviations in both the age and sea-level directions (where the standard deviation for each point in the sea-level reconstruction method is ± 6 m; Siddall *et al.*, 2003^[ref.13], 2004^[ref.14]). These probability intervals account for all of the combined uncertainties in both age and sea-level values and represent "worst case" propagation scenarios, given that no correlation was considered between any of the uncertainties. We then made $N=1000$ new records using independent random perturbations of all points within their probability distributions [Note 1: in the present study, $N=5000$]. [Note 2: A modification to that method in the present study is that we now apply an additional criterion, namely that there is strict stratigraphic coherence within the contiguously sampled KL11 record (allowing no age reversals during MC-resampling)]. This gives a 'sample' of [5000] RSL 'realisations' per equally spaced time step of 125 years [Note 3: this is close to the average time resolution of the original data, but not perfectly so, to avoid aliasing]. For each 'sample', we determine the 95% and 68% probability intervals for the entire distribution (from percentile counts) and the RSL value of the probability maximum [Note 4: this is the modal value in each pdf at each 125-y timestep, and we also analyse each pdf to determine the 95% confidence interval within which this modal value can be determined], which provides the 'best fit' RSL curve from the Red Sea data [with its 95% probability interval]. [Note 5: in the present study, we also determined the medians (50th percentile, for comparison with the probability maxima].

Because of the stratigraphic coherence in the record, we find that the modal value (and median) in each 125-y pdf of MC resamplings is tightly constrained, with the mode being typically constrained with 95% bounds of only ± 2 to 2.5 m. In the Grant *et al.* study, for the Red Sea stack, this was ± 6 m, because the stack does not have strict stratigraphic coherence from one datapoint to the next, so that relative age uncertainties between datapoints remained much larger than in our new record. Stratigraphic coherence in KL11 revealed a coherent LIG pattern (validated by KL23), and analyses that take these stratigraphic constraints into account resulted in a much more coherent and tightly defined record. This is simply because: (1) the initial (relative) age uncertainties were much smaller than in the Grant *et al.* study; (2) the record is coherent throughout rather than jumping between values from different, stacked

records as was the case in Grant *et al.* So, it makes sense that the probability maximum, median, and population percentiles all end up to be tighter-defined than in that study.

We have added some more text to the *Methods* to better clarify these points, but have not gone into full repetition of the Grant *et al* arguments, because this would be redundant.

When the reviewer asks “how confident we can be”, then the answer comes directly from the statistics: the 95% interval for the modal values means that if the MC exercise were run again, and again, and again, etc., 1 in 20 of the exercises might give modal values outside the envelope, while the other 19 exercises would return modal values within the range. We consider that to be rather logical, and not something that needs to be spelled out in the manuscript.

On the comment about the sea-level contributions from Greenland, which are up to ~5 m (4.1 to 6.2 m 95% credible intervals at 127 ka; Yau *et al.*, 2016^{ref.15}), relative to which the reviewer finds our +/-2 to 2.5 m 95% probability intervals “somewhat limiting for the residual calculations,” we emphasize that we have rigorously propagated ALL uncertainties into the final solutions (*Figure 3*), and that we then ONLY discuss signals that exceed the final 95% envelopes. Thus, we focus only on robust signals relative to all the limitations of the arguments, including those of the reviewer. We believe that this is quite clear in the manuscript, so we don't see what else may be added to make it even clearer, without adding redundant text. Also, it must be said that +/- 2 to 2.5 m at 95% is very precise for palaeo-sealevel reconstructions. Of course, we would also like more precision, but it is what it is..... Either way, none of our conclusions are affected.

L137: wording “may need to be amended”...

Agreed – the use of the term “for the first time” was likely to give a wrong impression. We have accordingly removed it.

L141: discussion of intra-LIG fluctuations and the importance of different reconstructions

The Kopp *et al.* (2013)^{ref.24} analysis (K13) built on the K09 study, adding additional constraints (limiting analysis to 129 to 116 ka) and expanding the suite of sea-level observations, and resolved at least one “sizeable intra-LIG sea level fall and rise, likely one in excess of 4 m. Moreover, the sea level rise following the lowstand occurred at a maximum kyr-averaged rate that likely exceeded 3 m kyr⁻¹, but was unlikely to have exceeded 7 m kyr⁻¹ and extremely unlikely (5 per cent probability) to have exceeded 11 m kyr⁻¹.” (we have added a statement to this effect) This structure and rates of rise compares favourably to our probabilistic assessment of KL11 (the rate of sea-level rise following the pronounced sea-level drop at ~125 ka is 0.8 m/century; *Figure 2g*). All K13 experiments (bar the ‘only Red Sea’ subset) had a ≥95 % probability of two peaks and a maximum intra-LIG rate of sea-level rise “likely exceeding 3.8 to 5.2 m kyr⁻¹, unlikely exceeding 6.1 to 7.6 m kyr⁻¹ and extremely unlikely exceeding 9.5 to 10.9 m kyr⁻¹”. In comparison, our new probabilistic assessment suggests the fastest rate of sea-level rise occurs at ~127 ka (~3.5 m/century) and the lowest rate of rise occurs after the pronounced sea-level fall (0.8 m/century). We have added some text to reflect the similarities, but also outline the differences between statistical compilations on arbitrary timescales (which will always smooth the dataset and thus suppress rate-of-change values) vs. comparisons between stratigraphically coherent, high-resolution records with strong independent age constraints. In addition, we emphasize in the manuscript that kilo-year averages (Kopp) are likely to hide larger centennial-scale averages (our study)

L171: Southern Ocean uranium mass accumulation rate changes

The spike in authigenic uranium at ~127 ka in Southern Ocean sediments is thought to have been caused by coastal freshening due to mass loss from the Antarctic ice sheet²⁶. Given the position of the core (ODP 1094, South Atlantic), the authigenic uranium mass accumulation rate (aU MAR) record may be somewhat site-specific, and the likely source may then be the West Antarctic Ice Sheet (WAIS). The lack of later spikes could then suggest either that the majority of the WAIS was lost (or at least was

significantly retreated inland) during the early portion of the LIG^{e.g., 27,28}. For example, Steig *et al.* (2015^{ref.29}) propose that significant atmospheric changes (hence temperature/isotope anomalies) consistent with the loss of the WAIS (which had mostly melted by ~126 ka) can account for the difference in the isotopic ($\delta^{18}\text{O}$) records from Mt. Moulton (west Antarctica) and East Antarctic ice core sites (Talos Dome, EDML, Vostok, Dome Fuji and EDC). Subsequent, water-isotope enabled climate simulations (Holloway *et al.*, 2016^{ref.30}) also suggest that the loss of WAIS (between 128 and 125 ka) AND a meltwater driven build-up of Southern hemisphere sea ice is needed to explain the divergent $\delta^{18}\text{O}$ signals observed in Antarctic ice cores. Further, Holloway *et al.* (2016), suggest that the largest differences (after ~126 ka) between the (persistently) low isotope values in the Mt. Moulton record and those in other ice core sites coincide with maximum retreat of the WAIS³⁰. Other ice core evidence for WAIS retreat/collapse comes from fragmentary blue-ice deposits. A hiatus in a blue-ice record from the Patriot Hills (located on the periphery of the WAIS) is also thought to indicate substantial LIG WAIS collapse, with these authors attributing retreat to ocean warming and destabilisation of (local) subglacial methane hydrates (Turney *et al.*, pre-print, submitted to PNAS³¹). This major ice mass loss from Antarctica seems to be corroborated by other emerging isotopic evidence (Sr-Nd-Pb isotopes of silt) from ODP Site 1096 in the Bellingshausen Sea, suggesting an absence of the WAIS during the LIG^{32,33}. Reconstructions of the East Antarctic Ice Sheet LIG elevations are even more uncertain than for the WAIS, with evidence suggesting either EAIS thinning (due to increased melting; e.g., Holden *et al.*, 2010^{ref.28}) or thickening (due to increased accumulation; e.g., Masson Delmotte *et al.*, 2010^{ref.34}, Holloway *et al.*, 2017^{ref.35}). Recent modelling suggests that the combined effects of EAIS lowering and meltwater forcing increased the proportion of cold season precipitation during the LIG, and hence could also go towards reconciling the divergent east vs. west early-LIG ice-core isotope observations (Holloway *et al.*, 2018^{ref.36}). However, isostatically driven changes in surface elevation alone currently fail to generate sufficient elevation-driven δD signals in east Antarctica (i.e., EAIS isostatic response to WAIS collapse is “not large”; Bradley *et al.*, 2012^{ref.37}).

In addition, the younger AIS melt periods in our study are much less sharp, with much lower rates of change, and could thus be sufficiently diluted in the dynamic Southern Ocean so AABW formation was not structurally inhibited. **The palaeoceanographic discussion is now part of the main manuscript, and we specifically address the question about minor, younger AIS meltwater influences.**

L181: role of ocean versus atmospheric forcing of AIS

At present, a key process driving mass loss from the Antarctic ice sheets is oceanic warming, i.e., intrusion of warm ocean currents into sub-shelf cavities^{e.g.,38}. These ice sheets are also vulnerable to changes in atmospheric conditions on their upper surfaces (e.g., through increased surface melting and increased hydro-fracturing, changes in accumulation etc.)^{e.g.,39,40}. For example, atmospheric warming of the Antarctic Peninsula, and associated increase in surface melting caused the loss of both Larsen A and Larsen B ice shelves, and dominates the thinning of the Larsen C ice shelf^{e.g.,38,41}.

At line 181 to 183, we suggest that both atmospheric and oceanic processes contributed to the early LIG Antarctic mass loss. The reviewer is correct, the difference in atmospheric CO_2 during the LIG is modest compared to today. However, the contrast between atmospheric CO_2 levels and Antarctic temperatures (calculated from the ice isotopes) in the early LIG compared to the preceding glacial period is large, which would imply that AIS experienced sustained atmospheric warming, that likely acted in tandem with oceanic warming. The atmospheric warming is not likely caused by CO_2 , but by heat release from the ocean.

We agree that our text was confusing, and have rephrased this line to remove this source of confusion.

L189: removal of reference to GrIS

We have clarified this statement and added additional references (King *et al.*, 2018^{ref.42}, van den Broeke *et al.*, 2016^{ref.43}).

L192: calculation of rate of ice build-up during the LIG

The reviewer states that AIS “*must be assumed as largely marine based during the LIG, with considerable calving*”. Unfortunately, this seems to be an opinion rather than something underpinned by references. We respectfully disagree that this necessarily would have been the case throughout the LIG, for two reasons:

- 1) In major retreat phases, there is competition between mass loss/retreat of marine margins and/or isostatic rebound (e.g., Gomez studies cited in our manuscript). The GIA feedbacks in these studies strongly slow ice sheet mass loss/retreat through rebound and re-grounding (e.g., Bradley *et al.*, 2015^{ref.44}; Gomez *et al.*, 2013, 2015^{refs.45,46}; Konrad *et al.*, 2015^{ref.47}). Also during the warm Holocene, extensive climatically driven retreat was halted by glacio-isostatic rebound-induced re-advance of the WAIS (Kingslake *et al.*, 2018⁴⁸).
- 2) Accumulation was simply held constant in our calculations, but with the documented warmer sea surface temperatures (SSTs) and (up to 4°C) warmer Antarctic air temperatures, more moisture would have been available (up to ~30% extra, using the Clausius-Clapeyron relationship for $\Delta T=4^\circ\text{C}$; and, in agreement with that estimate, accumulation rates at EPICA Dome C went up from ~30 to ~39 kg/m⁻²y⁻¹ through this event [Figure 8 of Wolff *et al.*, 2010^{ref.49}]). Also, lower AIS topography has been linked with intensified cyclones over the continent, which further enhances the potential for increased accumulation (Walsch *et al.*, 2000^{ref.50}, Holloway *et al.*, 2016^{ref.30}). We have added some text to this effect in both the main text, and the *Supplementary Information*. Our calculations are clearly stated to be ball-park approaches only.

L199: rates of sea-level rise

This is a comment of our record (coherent, centennial resolution) versus K13 (statistical compilation). It has been dealt with above, and we have brought out this kilo-year vs. centennial-scale difference in the main text.

About the differences with the DeConto and Pollard (2016^{ref.20}) study highlighted by the reviewer: in our view, the calculated high rates ARE found in the early LIG deglaciation of the AIS (except that there is an interruption in that deglaciation, likely due the GIA rebound and ice regrounding). But also, DeConto and Pollard’s parameterization is the first approach to including such processes (we will not comment on the reviewers ‘unphysical’ argument); it’s not perfect, but at least it’s a step in the direction of understanding fast rates of sea-level rise, which we know have happened at other times in the past and now possibly also during the LIG. Recent analysis of iceberg keel plough marks from Pine Island Bay (Wise *et al.*, 2017^{ref.51}) suggest that during the last deglaciation calving-margin thicknesses were equivalent to the threshold that is predicted to trigger ice-cliff structural collapse and these authors infer rapid and sustained ice-sheet retreat driven by the marine ice-cliff instability processes (cf. Pollard *et al.*, 2015^{ref.52}). We think it would be too much of a tangent to discuss this in the manuscript – it is in our opinion more of a target for ice-sheet modelling than for sea-level reconstruction from observations.

L301: linear interpolation of Red Sea record during the LIG period

Yes, the reviewer is almost correct, other than that we did this for the age model with (non-linear) Hermite splines in a Monte Carlo style approach, adhering to stratigraphic constraints to ensure that the MC method does not introduce spurious age reversals, and then for the RSL record using linear interpolations between points. RSL interpolations are made AFTER placing the record in the time/age domain. The RSL interpolations, therefore, are linear interpolations in the age domain, but non-linear interpolations in the depth domain. We believe that the more expanded information on the Red Sea RSL method in the *Methods* section now presents sufficient information for this to be clear.

Figure 3 shading: the shading was/is explained already, in the caption.

REVIEWER 2

1. “They claim this reconstruction is stratigraphically “stronger” than those derived from reefs, yet fail to state uncertainties”

This seems to be a confusion. We actually stated that:

- a. “stratigraphic **coherence** and, therefore, **relative age relationships are stronger**” (lines 50 to 51 in the original manuscript) (to clarify this better, we have elaborated on the stratigraphic coherence of Red Sea sediment cores in the *Methods* section, lines 423-436)
- b. the total propagated vertical uncertainty associated with the probability maximum sea-level reconstruction that we present is ± 2.0 to 2.5 m at 95 % probability (line 129 in the original manuscript; now in line 161). We have now added details on how this was determined relative to the uncertainties in individual data and in the previous Red Sea stack are given in the *Methods* section, lines 471-492.

2. Balance (or reviewer perceived lack thereof) in representing/synthesising available LIG sea-level data; “confirmation bias” and the “uncritical acceptance” of the Seychelles and Bahamas coral sea-level records

We are a bit surprised by this comment, given that we clearly described and acknowledged the ongoing debate regarding the nature (rate, amplitude etc.) and number/existence of sea-level oscillations during the LIG (lines 25-26, 43-45, 75-76, 154 and 313 in the original manuscript), the divergent and often contradictory field evidence^{e.g.,57} (lines 74 to 75, 313 to 322 and in section 1 of the original *Supplementary Information*), and that we presented extensive stratigraphic evidence for LIG sea-level oscillations from different archives (e.g., corals, sedimentary sequences) for **>20 sites** (which filled 12 pages of the original *Supplementary Information* section 1). Hence, we think that we did extensively demonstrate that there is substantial debate, and that we’re not hiding that at all. Note that **reviewer 1**, in contrast, was actually highly complimentary about the **“wealth of details in the *Supplementary Information* that make it possible to follow the argumentation.”**

Regardless, we have now drawn attention to the debate more directly in lines 80-90. Moreover, we added another (very recent) site study to *Supplementary Information* section 1, about West Caicos.

In addition, we summarised the key benefits and limitations of both coral-based reconstructions and the sediment-core-based Red Sea record in the second paragraph (lines 47–59 in the original manuscript; now lines 52-64 in the revised manuscript). For even more thorough review of the uncertainties, limitations, benefits, and also the opportunities associated with all different methods of reconstruction past sea levels (e.g., coral, salt-marsh, sediment cores), we refer to several excellent books and review papers that are readily available already (e.g., van de Plassche, 1986^{ref.58}, Shennan *et al.*, 2015^{ref.59}, Montaggioni, 2005^{ref.60}).

The reviewer then contends that “*uncritical acceptance of these [Seychelles and Bahamas] data, ... does them no favours and suggests confirmation bias*”, and that we “*claim Seychelles and Bahamas coral records “represent the ‘best sites’ for comparison”*. We find this in conflict with the extensive assessment that we actually provided in the *Supplementary Information*. It is true that we did bring the specific Seychelles and Bahamas records out our synthesis of *Supplementary Information* section 1, but only after they were assessed along with all the others (see also *Figure 1*, and *Supplementary Figure S3*). It is quite clear from those representations that there is no one single, perfect LIG sea-level site, as we now specifically highlight in lines 87-90), but that there is an emerging pattern. We have brought the selection criteria for the Seychelles and Bahamas much stronger to the fore in *Supplementary Information* section 1c, as: “*The intensively studied, sampled, and dated LIG coral/reef records of the Seychelles⁸⁻¹⁰, Bahamas^{15,19}, and Western Australia^{18,69-73} give an emerging picture of LIG sea level. These records are especially useful given that: (1) they span extended periods of the LIG, (2) they have relatively*

high temporal sampling and density of radiometric dating, (3) they are from tectonically stable areas; (4) they have well-documented stratigraphic superposition of LIG units, and (5) for the Seychelles, there are well-constrained palaeo-water depth estimates. We do not view these records in isolation, but within the well-documented context of the records extensively discussed in sections 1A and 1B.” (Refs in this cited section are from the actual *Supplement*)

A few other potential sites fulfil our criteria of the combined detailed stratigraphy **and** large number of U-series ages; e.g., the Yucatan, Western Australia and Barbados. We did not use the well-dated and stratigraphically well-characterised Yucatan record (Blanchon *et al.*, 2009^{ref.62}) because this is a back-stepping reef sequence (i.e., no superposition of LIG reef units, and therefore no strict stratigraphic control), and because the authors did not provide any information on the palaeo-water depth of the corals dated. The elevated reef sequences of Western Australia provide compelling evidence for higher than present sea levels (~+2 to +4 m above present)^{63–67}, with some suggestion of a late interglacial sea level of ~+9 m (from a mapped shoreline at +5 to +6 m; O’Leary *et al.*, 2013, 2008b^{refs.19,64}). However, the absolute age and sequence of events, particularly the late LIG highstand in Western Australia, remains controversial. For example, some U-series ages for the LIG in the Australian region may suffer from remobilisation of nuclides (open-system behaviour) and/or post-depositional chemical alteration (diagenesis) of the coral terraces (e.g., Stirling *et al.*, 1998, 1995^{refs.65,66}). Likewise, LIG sea-level estimates from the Quobba Ridge (+9 m after GIA correction¹⁹) may be affected by tectonic deformation⁶⁸ [note also that a recent LIG GIA fingerprinting exercise (Hay *et al.*, 2014^{ref.69}) demonstrated a closer accord between the Seychelles and Western Australia sea-level data following GIA correction]. The Barbados record offers some exquisite 3-dimensional LIG reef sequences, but evidence for LIG sea-level instability is equivocal, and the interpretation of Barbados reefs is complicated by both tectonics and GIA processes (e.g., Austermann *et al.*, 2013^{ref.70}). While not brought into the synthesis, these records are all discussed in detail in the *Supplementary Information*.

In view of the above, and in the absence of peer-reviewed literature (that we could find) concerning the objections that the reviewer raises about some of these sites, we maintain that the Seychelles and the Bahamas are among the best to compare with. More importantly, our argument does not hang on the similarity of the Seychelles and Bahamas coral records to the Red Sea sea-level reconstruction, especially given also the palaeoceanographic support, which the reviewer seems to have overlooked (main text *Figure 3*; *Supplementary Information* section 6). To ensure that the palaeoceanographic evidence is considered with equal weight, we have brought that entire suite of material out of the *Supplement* and into the main paper (separate section in *Discussion*, including *Figure 4*).

Further, in response to specific comments the reviewer made about the Seychelles and Bahamas:

2.1. Seychelles coral sea-level record

The reviewer raises several concerns relating to the Dutton *et al.* (2015) Seychelles record. Although these might better be addressed to the authors of that study, e.g., through a Comment, we here address our views on these concerns.

(a) in situ vs clasts:

We have used all the available information **as published**, as long as no **published objections are available**. The samples used in our comparison, (*Figure 2f* and *S3*) were clearly described as *in situ* samples, and they pass our age reliability screening (we use them as an inverse weighted mean for replicate analyses).

- Montaggioni and Hoang, 1988^{ref.71}: 5 alpha-counting U-series dates undertaken on “*in situ* coral colonies”. Note, these alpha dates are not included in our synthesis due to their large analytical error (± 10 ka).

- Israelson and Wolfarth, 1999^{ref.72}: “all coral colonies sampled for analyses are in **growth position**”
- Dutton *et al.*, 2015: “We undertook detailed surveying, sedimentologic description, and sampling of outcrops that displayed **in situ** coralgall framework at several sites on both La Digue and Curieuse islands”; “Vertical successions of **in situ** fossil corals from the Seychelles record [show] a gradual sea-level rise between ~129 and 125 ka. An intervening layer of coral rubble just before 125 ka in two outcrops indicates that this gradual rise may have been briefly interrupted, but the meaning of this rubble layer is still open to interpretation”
- Vyverberg *et al.*, 2018: these authors build on the outcrop descriptions of Dutton *et al.*, 2015. “While some of the outcrops are entirely composed of cemented reef rubble, the others that we focused on consist predominately of **in situ** coralgall framework and cemented coralgall rubble”, “We targeted **in situ** coral and associated biota from each lithostratigraphic unit for sampling.”

In sum, **the reviewer’s concerns about working with loose coral clasts seem unjustified in the context of the material we focussed on.** Hence, we have decided not to add any text to this effect to the manuscript, because it would only lead to confusion and because it really is tangential to the actual scope of our study.

(b) non-replication of intra-sample ages:

This also seems unjustified. All new analyses presented in Dutton *et al.* (2015) were clearly described as dated in triplicate to assess both reproducibility and any diagenetic trends. Current screening methods (% calcite, $\delta^{234}\text{U}_{\text{initial}}$, ^{232}Th concentration etc.) do not always fully account for all possible combinations of loss or gain of nuclides and replicate dating is a useful means of assessing any open system behaviour. The authors report that four corals have “unusually large intra-sample variability in ages”; one of these had elevated [^{232}Th], two had concentrations suggestive of U-addition. All four of those suspect samples were removed both from further analysis in Dutton *et al.* (2015), and also from our summary figures.

(c) fails to consider stratigraphic consistency of ages and rates of framework development:

There are no straightforward stratigraphic relationships between individual coral colonies. Unlike sediment layers, coral reefs grow in 3-dimensions, so that a prograding reef may contain a wide range of ages along the same elevation (or terrace), and these may be interlocked in a complex 3-dimensional manner. The implication is that data cannot be rejected where multiple ages are observed for a single elevation (as opposed to for a single coral sample). Likewise, age inversions that are identified on the basis of age and elevation data alone cannot be rejected out of hand because such relationships may be consistent with 3-dimensional development of the reef.

With respect to the comment that coral ages should vary by less than 1,000 years per unit, we emphasize that this value was obtained for a specific reef in one specific location (Huon Peninsula, Papua New Guinea), where the total age range per horizon was found to be 800–1060 year (Edinger *et al.*, 2007^{ref.73}). It remains undetermined if that may be projected to other regions, given that each reef is its own complex system that comprises primary and secondary growth frameworks, marine cementation, mechanical and biological erosion, and post-depositional diagenesis. It is widely acknowledged that, for both modern death assemblages and fossil assemblages, understanding of the time-averaging taphonomic processes “of corals and other non-molluscan reef groups [...] is still nascent” (Kidwell, 2013^{ref.74}).

In the 3 new sites presented by Dutton *et al.* (2015), samples were taken from different units of the described stratigraphic sequence (sites 4, 7 and 19A). For each of these sites, the U-series ages of the samples **are** in stratigraphic order (the samples SY36 and SY37 are *prima facie* out of

sequence but these two dates overlap within uncertainty and are very close in elevation, and can thus not be statistically or stratigraphically distinguished).

The reviewer then contends that the rates of framework development in the Seychelles record of Dutton *et al.* (2015) are inconsistent with those described for general reef framework development, which (s)he relates to Holocene data from the Huon Peninsula study. First, we again question whether insights from specific locations may be applied to other locations. Reef frameworks tend to accrete much more slowly (0.1 to 1 cm/yr) than individual coral colonies (~1 to 10 cm/yr; Hopley *et al.*, 2007^{ref.75}, and also *Supplementary Information* section 5). More importantly, no generalisations can be made because substrate availability, water depth, energy and trophic conditions are key factors that determine accretion rates (Camoin and Webster, 2015^{ref.76}, Montaggioni, 2005^{ref.60}; Woodroffe and Webster, 2014^{ref.77}), and the factors are predominantly **locally** controlled. For Holocene reef sequences in the Seychelles, irregularities in age/depth distributions “imply an irregular surface with metre-scale relief and little accretion, or that there was local active erosion, generating relief and recycling older material”, and age irregularities that are possible evidence of “diagenetic alteration and contamination by fresh water” (Braithwaite *et al.*, 2000⁷⁸). These authors concluded that the distribution of dates (U-series) was “hard to explain in terms of a progressive accretion of the reef surface, but is entirely in keeping with interpretations based on exceptional erosion and recycling of material” (Braithwaite *et al.*, 2000⁷⁸). Rates of reef accretion from this site (reef edge, Anse aux Pins) are ~ 0.023 cm/ka, which is very similar to those estimated for the three Dutton *et al.* (2015) LIG sites (0.013 to 0.03 cm/yr) and less than the maximum average Holocene accretion rates for the Mauritius and Seychelles (0.031 cm/yr; Camoin and Webster, 2015).

The outcrops/sampling strategy of Dutton *et al.* (2015) does not permit a holistic examination of the accretion rate of the Seychelles LIG reefs (as the youngest portion appears to be ‘missing’). However, for the three sections mentioned above (sites 4, 7 and 11), rates of accretion varied from 0.13 to 0.3 m/ka (0.013 to 0.003 cm/yr). Such rates are in line with Holocene reconstructions (above) and with average accretion rates for exposed and semi-exposed/sheltered reef settings in the Indo-Pacific (0.15 to 1.2 cm/yr and 0.1 to 2.5 cm/yr, respectively) (Montaggioni, 2005^{ref.60}). Overall, these comments would be better addressed to the original authors (and their reviewers) and we have decided not to add any further text on these tangents to our arguments to the *Supplementary Information*, especially given that we were given no specific references to substantiate some of the concerns that were voiced.

(d) Southern Ocean swell and “coralgal development above mean SL”

No additional references were given for this comment, but in view of the position of the Seychelles close to the equator, Southern Ocean swells seem unlikely from an oceanographic point of view. Without references or concrete information, we could not address a point framed like this.

In any case, the modern reef assemblages of Mahé (Seychelles; Taylor, 1968), prior to an extensive El Niño bleaching event which reduced species diversity (Goreau, 1998), indicates that for most shores (other than very exposed) corals and coralline algae occur below mean low water neap (MLWN) and the majority below mean low water springs (MLWS), so NOT up to 2 m above mean sea level as stated by the reviewer (which, incidentally, likely is a value gleaned from Tahiti; Cabioch *et al.*, 1999—and its application to the Seychelles is questionable). Algal ridges on Mahé are restricted to windward and partially exposed reefs and can extend 0.5 m above the reef flat, which is still within the eulittoral zone (Taylor, 1968). Most importantly, **“the shallow waters around the granitic [Seychelles] islands provide an effective barrier to long-period swell waves”**

(Braithwaite *et al.*, 2000^{ref.78}). We have not added any text about this because we feel that it is a confusing tangent with limited relevance to the sites discussed.

2.2. Bahamas coral sea-level record

When the reviewer states “the same is true for the Bahamas, despite good data to contest their findings”, then it is not clear what “the same” stands for? Surely not Southern Ocean swell. The comment is simply too vague for us to be able to address it. Also, the “good data contesting their findings” part of the comment is unsubstantiated. It gives us nothing to work with because no references are given, and it is insufficiently clear which of our findings these data would contest. We stress that the Bahamas one of the more complete records around. It has well-documented stratigraphy (two superimposed LIG reef units), which stands even without any datings. But in addition, there are lots of high-quality ages. New work from the region again confirms a massive intra-LIG unconformity, exposed over a 5km distance (nearby West Caicos; Kerans *et al.*, 2019^{ref.79}).

The latter has been added to *Supplementary Information* section 1.

3. The 2-4 m ‘discrepancy’ between the Red Sea record and the coral records from the Bahamas and Seychelles; and our failure to discuss the Red Sea coral record in view of neotectonics.

First, we refer clearly to the source papers for the range of 5 to 10 m from Red Sea reefs. The source study actually deals with the neotectonics (as correctly mentioned by the reviewer), and **the final range for sea level that we obtain from the study is based on the conclusion of Plaziat *et al.* themselves** (the source study), which they came to **after taking the various deformations into account**. It would therefore be misplaced for us to make such corrections again because that would distort the careful work represented in the source studies. Second, the higher than “normal” (which the reviewer does not specify) position of LIG corals in the Red Sea may be exactly because (neo-)tectonics might be needed to allow registration of the highest phases of the LIG, as we have outlined in *Supplementary Information* section 5, and **also in main text lines 193-198**). Note also that the LIG sea-level compilations of Kopp *et al.* (2009 and 2013^{refs.22,24}) emphasized very similar ranges as the Plaziat⁸⁰⁻⁸² and Bruggemann⁸⁵/Walter⁸⁴ studies, so the “higher than normal” comment does not seem justified in view of Kopp’s recent global compilations.

4. Balance (or reviewer perceived lack thereof) in representing/synthesising available LIG sea-level data and possible refocusing of the manuscript.

Thanks—We can see where this perception came from (we relied too much on the *Supplement*), and we have accordingly added text to the main manuscript (and the caption of *Figure 2*) to make this clearer.

On the comment that the paper may focus too much on the reversal around 124 ka, we disagree. It is an important (but not exclusive) part of the comparison with coral records, but our paper is much broader than just the coral records (that make up <20% of the study). The actual focus is on unravelling the timing relationship between the Antarctic and Greenland ice-mass histories in a statistically relevant manner with full propagation of all uncertainties, and then on comparing this with key palaeoceanographic data. The clearly crucial separation for the AIS vs. GrlS story resides close to 125-124 ka, which likely is where the impression comes from that we are only interested in that. But the reviewer ignores that we do actually discuss THREE meltwater events (R1, R2 and R3). We are not yet in a position to discuss the nature of the potential “R4” at ~120ka because there is less agreement about its sea-level position (see *Figure 2f, g*). For Antarctica, our reconstruction suggests that it is an event that is already within the general accretion phase, when variations will be differently controlled than during “deglaciation” phases. Our paper is more focussed on the LIG highstand and how it came about.

1. Presentation of the KL23& KL11

Thanks – We have adjusted the graph to make the smoothings much less prominent.

2. Replicate data (KL23)

This is the nature of replication. It is perhaps not nice to see, but it is what comes out of repeated analyses. Note that the uncertainty on the grey box is small because it is the Standard Error of the mean: SD/\sqrt{N} . The standard deviation (SD) on each of the replicate RSL values is portrayed by the blue cross. Hence, it is obvious that individual measurements may drift quite a bit whereas a mean of several replicates is much more constrained. Still, we do not wish to show this interval using ONLY the means because the red values were all measured in one go one mass spectrometer, whereas the grey values represent analyses on another mass spec more than a year later. The *a-priori* comparability between red and red is therefore more certain than comparability between red and grey.

Given that the SD cross is shown and that the grey values were described in the caption as means with SE bars, we think this was sufficiently clarified already, and left the text as it was.

3. GIA and Yucatan speleothem record

The reviewer is correct. This was a clear oversight; thanks for spotting it.

We have addressed this by adding this, with considerable elaboration and including Figure 1 below, to *Supplementary Information* section 2.

Specifically, we have corrected both the Red Sea record and the Yucatan Peninsula speleothem record for GIA processes using adjusted (more realistic) configurations of the penultimate glacial (MIS 6) ice sheets (cf. Rohling *et al.*, 2017^{ref.9}). This uses a smaller Laurentide Ice Sheet with either: (i) a Eurasian Ice Sheet (EIS) with greater mass but LGM-like spatial configuration; or (ii) an EIS with both greater mass and spatial extent. Fuller details of the chosen Earth model and ice models are given in *Supplementary Information* section 4, where it is shown that the GIA corrections themselves have uncertainties up to ± 3 m at the end of the LIG. The exercise used here for evaluating the Red Sea versus Yucatan record after GIA correction uses an artificially defined Global Mean Sea Level (ice-volume) history. It shows that the Yucatan record closely tracks GMSL. The Yucatan data used in *Supplementary Figure S3* indicate that Yucatan RSL (and thus by close approximation GMSL) first reach -4.9 m after just after about 118 ka. The upper 95 % confidence bound for Red Sea RSL would sit some 3 m above that (See Figure 1 below), with an uncertainty up to ± 3 m (*Supplementary Figure S4*); hence our selection of the ~ 118.5 ka age for the upper 95% bound of the Red Sea record to fall through 0 m. Bearing in mind the generous 2σ (95%) uncertainty of ± 1.2 ka that applies to the Red Sea age model (*Supplementary Figure S2*), this selection of ~ 118.5 ka is coherent with both the Yucatan data and the Cutler *et al.* (2003)⁸⁵ and Hibbert *et al.* (2016)⁸⁶ assessments for the end of the LIG.

Figure 1: GIA predictions of relative sea level for the Red Sea stack (solid red line = median; dashed red line = upper 95 % confidence interval), Yucatan Peninsula (blue) and global mean sea level (GMSL, black) using 'more realistic' MIS 6 ice histories. We use a VM-2-like earth model, a smaller volume Laurentide Ice Sheet and: (a) greater volume Eurasian Ice Sheet with LGM-like spatial configuration (ICE 3); and (b) greater volume Eurasian Ice Sheet with more extensive spatial extent (ICE 4).

4. L120-121 KL11 “best core”

Note that we actually state that KL11 is from the “best (central) location for Red Sea quantification”. The sea-level calculations (Siddall *et al.*, 2002, 2004^{refs.87,88}) are focused on the central Red Sea (18 to 20 °N). The northern sector of the basin has more complex oceanography (i.e., overturning and the formation of Red Sea Deep Water), and also some influence of Mediterranean weather systems. Siddall also gave tentative calculations for the northern Red Sea (worked out to approx. a linear variation of $\delta^{18}\text{O}_{\text{calcite}}$ with distance from the Hanish Sill), which does give a reasonable sea level solution for $\delta^{18}\text{O}_{\text{calcite}}$ records at the latitude of KL23, but they are less well constrained than for the central sector. The central Red Sea cores are also more suitable than northern/southern cores because they are not subject to variability linked to Mediterranean weather influences, or (as is the case in the southern Red Sea) greater variability associated with seasonal incursion of Gulf of Aden Intermediate Water.

We have added some text to clarify this – lines 148-153.

5. L137-139 Remove “for the first time”

Agreed – removed.

6. L348 authors meant Figure 2e

Good spot - thank you. Yes, we did mean Figure 2e and have corrected this in the manuscript. We have carefully cross-referenced all figure calls in the revised manuscript and supplement.

References cited:

1. McKay, N. P., Overpeck, J. T. & Otto-Bliesner, B. L. The role of ocean thermal expansion in Last Interglacial sea level rise. *Geophys. Res. Lett.* **38**, L14605 (2011).
2. Hoffman, J. S., Clark, P. U., Parnell, A. C. & He, F. Regional and global sea-surface temperatures during the

- last interglaciation. *Science*. **355**, 276–279 (2017).
3. Jacob, T., Wahr, J., Pfeffer, W. T. & Swenson, S. Recent contributions of glaciers and ice caps to sea level rise. *Nature* **482**, 514–518 (2012).
 4. Radić, V. & Hock, R. Regional and global volumes of glaciers derived from statistical upscaling of glacier inventory data. *J. Geophys. Res.* **115**, F01010 (2010).
 5. Farinotti, D. *et al.* A consensus estimate for the ice thickness distribution of all glaciers on Earth. *Nat. Geosci.* **12**, 168–173 (2019).
 6. Milne, G. A. & Mitrovica, J. X. Searching for eustasy in deglacial sea-level histories. *Quat. Sci. Rev.* **27**, 2292–2302 (2008).
 7. Williams, F. H. A geophysical approach to reconstructing past global mean sea levels using highly resolved sea-level records. (University of Southampton, 2016).
 8. Lambeck, K. *et al.* Sea level and shoreline reconstructions for the Red Sea: isostatic and tectonic considerations and implications for hominin migration out of Africa. *Quat. Sci. Rev.* **30**, 3542–3574 (2011).
 9. Rohling, E. J. *et al.* Differences between the last two glacial maxima and implications for ice-sheet, $\delta^{18}\text{O}$, and sea-level reconstructions. *Quat. Sci. Rev.* **176**, 1–28 (2017).
 10. Koerner, R. M. & Fisher, D. A. Ice-core evidence for widespread Arctic glacier retreat in the Last Interglacial and the early Holocene. *Ann. Glaciol.* **35**, 19–24 (2002).
 11. Zdanowicz, C. M., Fisher, D. A., Clark, I. & Lacelle, D. An ice-marginal $\delta^{18}\text{O}$ record from Barnes Ice Cap, Baffin Island, Canada. *Ann. Glaciol.* **35**, 145–149 (2002).
 12. Grant, K. M. *et al.* Rapid coupling between ice volume and polar temperature over the past 150,000 years. *Nature* **491**, 744–747 (2012).
 13. Siddall, M. *et al.* Sea-level fluctuations during the last glacial cycle. *Nature* **423**, 853–858 (2003).
 14. Siddall, M. *et al.* Understanding the Red Sea response to sea level. *Earth Planet. Sci. Lett.* **225**, 421–434 (2004).
 15. Yau, A. M., Bender, M. L., Robinson, A. & Brook, E. J. Reconstructing the last interglacial at Summit, Greenland: Insights from GISP2. *Proc. Natl. Acad. Sci.* **113**, 9710–9715 (2016).
 16. Cuffey, K. M. & Marshall, S. J. Substantial contribution to sea-level rise during the last interglacial from the Greenland ice sheet. *Nature* **404**, 591–594 (2000).
 17. Overpeck, J. T. *et al.* Paleoclimatic Evidence for Future Ice-Sheet Instability and Rapid Sea-Level Rise. *Science*. **311**, 1747–1750 (2006).
 18. Dutton, A. & Lambeck, K. Ice volume and sea level during the last interglacial. *Science*. **337**, 216–219 (2012).
 19. O’Leary, M. J. *et al.* Ice sheet collapse following a prolonged period of stable sea level during the last interglacial. *Nat. Geosci.* **6**, 796–800 (2013).
 20. DeConto, R. M. & Pollard, D. Contribution of Antarctica to past and future sea-level rise. *Nature* **531**, 591–597 (2016).
 21. Dutton, A. *et al.* Sea-level rise due to polar ice-sheet mass loss during past warm periods. *Science*. **349**, 153–165 (2015).
 22. Kopp, R. E., Simons, F. J., Mitrovica, J. X., Maloof, A. C. & Oppenheimer, M. Probabilistic assessment of sea level during the last interglacial stage. *Nature* **462**, 863–867 (2009).
 23. Düsterhus, A., Tamisiea, M. E. & Jevrejeva, S. Estimating the sea level highstand during the last interglacial: a probabilistic massive ensemble approach. *Geophys. J. Int.* **206**, 900–920 (2016).
 24. Kopp, R. E., Simons, F. J., Mitrovica, J. X., Maloof, A. C. & Oppenheimer, M. A probabilistic assessment of sea level variations within the last interglacial stage. *Geophys. J. Int.* **193**, 711–716 (2013).
 25. Rohling, E. J. *et al.* High rates of sea-level rise during the last interglacial period. *Nat. Geosci.* **1**, 38–42 (2008).
 26. Hayes, C. T. *et al.* A stagnation event in the deep South Atlantic during the last interglacial period. *Science*. **346**, 1514–1517 (2014).
 27. Vaughan, D. G., Barnes, D. K. A., Fretwell, P. T. & Bingham, R. G. Potential seaways across West Antarctica. *Geochemistry, Geophys. Geosystems* **12**, n/a-n/a (2011).
 28. Holden, P. B. *et al.* Interhemispheric coupling, the West Antarctic Ice Sheet and warm Antarctic interglacials. *Clim. Past* **6**, 431–443 (2010).
 29. Steig, E. J. *et al.* Influence of West Antarctic Ice Sheet collapse on Antarctic surface climate. *Geophys. Res. Lett.* **42**, 4862–4868 (2015).
 30. Holloway, M. D. *et al.* Antarctic last interglacial isotope peak in response to sea ice retreat not ice-sheet

- collapse. *Nat. Commun.* **7**, 12293 (2016).
31. Turney, C. *et al.* Early Last Interglacial ocean warming drove substantial ice mass loss from Antarctica. (2018). doi:10.17605/OSF.IO/3Z69P
 32. Carlson, A. E. *et al.* Absence of the West Antarctic ice sheet during the last interglaciation. in *2018 AGU Fall Meeting* PP11A-05 (2018).
 33. Voosen, P. Discovery of recent Antarctic ice sheet collapse raises fears of a new global flood. *Science* (2018). doi:10.1126/science.aaw4182
 34. Masson-Delmotte, V. *et al.* EPICA Dome C record of glacial and interglacial intensities. *Quat. Sci. Rev.* **29**, 113–128 (2010).
 35. Holloway, M. D. *et al.* The Spatial Structure of the 128 ka Antarctic Sea Ice Minimum. *Geophys. Res. Lett.* **44**, 11,129–11,139 (2017).
 36. Holloway, M. D., Sime, L. C., Singarayer, J. S., Tindall, J. C. & Valdes, P. J. Simulating the 128-ka Antarctic Climate Response to Northern Hemisphere Ice Sheet Melting Using the Isotope-Enabled HadCM3. *Geophys. Res. Lett.* **45**, 11,921–11,929 (2018).
 37. Bradley, S. L., Siddall, M., Milne, G. A., Masson-Delmotte, V. & Wolff, E. W. Where might we find evidence of a Last Interglacial West Antarctic Ice Sheet collapse in Antarctic ice core records? *Glob. Planet. Change* **88–89**, 64–75 (2012).
 38. Pritchard, H. D. *et al.* Antarctic ice-sheet loss driven by basal melting of ice shelves. *Nature* **484**, 502–505 (2012).
 39. Scambos, T. A., Hulbe, C. & Fahnestock, M. Climate-induced ice shelf disintegration in the Antarctic Peninsula. in *Antarctic Peninsula Climate Variability: Historical and Paleoenvironmental Perspectives*, *Antarctic Research Series* 79 72–92 (2003).
 40. Gorodetskaya, I. V. *et al.* The role of atmospheric rivers in anomalous snow accumulation in East Antarctica. *Geophys. Res. Lett.* **41**, 6199–6206 (2014).
 41. van den Broeke, M. Strong surface melting preceded collapse of Antarctic Peninsula ice shelf. *Geophys. Res. Lett.* **32**, L12815 (2005).
 42. King, M. D. *et al.* Seasonal to decadal variability in ice discharge from the Greenland Ice Sheet. *Cryosph.* **12**, 3813–3825 (2018).
 43. van den Broeke, M. R. *et al.* On the recent contribution of the Greenland ice sheet to sea level change. *Cryosph.* **10**, 1933–1946 (2016).
 44. Bradley, S. L., Hindmarsh, R. C. A., Whitehouse, P. L., Bentley, M. J. & King, M. A. Low post-glacial rebound rates in the Weddell Sea due to Late Holocene ice-sheet readvance. *Earth Planet. Sci. Lett.* **413**, 79–89 (2015).
 45. Gomez, N., Pollard, D. & Mitrovica, J. X. A 3-D coupled ice sheet – sea level model applied to Antarctica through the last 40 ky. *Earth Planet. Sci. Lett.* **384**, 88–99 (2013).
 46. Gomez, N., Pollard, D. & Holland, D. Sea-level feedback lowers projections of future Antarctic Ice-Sheet mass loss. *Nat. Commun.* **6**, 8798 (2015).
 47. Konrad, H., Sasgen, I., Pollard, D. & Klemann, V. Potential of the solid-Earth response for limiting long-term West Antarctic Ice Sheet retreat in a warming climate. *Earth Planet. Sci. Lett.* **432**, 254–264 (2015).
 48. Kingslake, J. *et al.* Extensive retreat and re-advance of the West Antarctic Ice Sheet during the Holocene. *Nature* **558**, 430–434 (2018).
 49. Wolff, E. W. *et al.* Changes in environment over the last 800,000 years from chemical analysis of the EPICA Dome C ice core. *Quat. Sci. Rev.* **29**, 285–295 (2010).
 50. Walsh, K. J. ., Simmonds, I. & Collier, M. Sigma-coordinate calculation of topographically forced baroclinicity around Antarctica. *Dyn. Atmos. Ocean.* **33**, 1–29 (2000).
 51. Wise, M. G., Dowdeswell, J. A., Jakobsson, M. & Larter, R. D. Evidence of marine ice-cliff instability in Pine Island Bay from iceberg-keel plough marks. *Nature* **550**, 506–510 (2017).
 52. Pollard, D., DeConto, R. M. & Alley, R. B. Potential Antarctic Ice Sheet retreat driven by hydrofracturing and ice cliff failure. *Earth Planet. Sci. Lett.* **412**, 112–121 (2015).
 53. Rohling, E. J. *et al.* Magnitudes of sea-level lowstands of the past 500,000 years. *Nature* **394**, 162–165 (1998).
 54. Fenton, M. Late Quaternary history of Red Sea outflow. (University of Southampton, 1998).
 55. Roberts, A. P., Rohling, E. J., Grant, K. M., Larrasoana, J. C. & Liu, Q. Atmospheric dust variability from Arabia and China over the last 500,000 years. *Quat. Sci. Rev.* **30**, 3537–3541 (2011).
 56. Rohling, E. J., Grant, K. M., Roberts, A. P. & Larrasoana, J.-C. Paleoclimate Variability in the Mediterranean and Red Sea Regions during the Last 500,000 Years. *Curr. Anthropol.* **54**, S183–S201 (2013).

57. Long, A. J. *et al.* Near-field sea-level variability in northwest Europe and ice sheet stability during the last interglacial. *Quat. Sci. Rev.* **126**, 26–40 (2015).
58. van de Plassche, O. *Sea-level research : a manual for the collection and evaluation of data.* (Geo Books, 1986).
59. Shennan, I., Long, A. J. & Horton, B. P. *Handbook of sea-level research.* (John Wiley and Sons Ltd., 2015).
60. Montaggioni, L. F. History of Indo-Pacific coral reef systems since the last glaciation: Development patterns and controlling factors. *Earth-Science Rev.* **71**, 1–75 (2005).
61. Woodroffe, S. A., Long, A. J., Milne, G. A., Bryant, C. L. & Thomas, A. L. New constraints on late Holocene eustatic sea-level changes from Mahé, Seychelles. *Quat. Sci. Rev.* **115**, 1–16 (2015).
62. Blanchon, P., Eisenhauer, A., Fietzke, J. & Liebetrau, V. Rapid sea-level rise and reef back-stepping at the close of the last interglacial highstand. *Nature* **458**, 881–884 (2009).
63. Eisenhauer, A., Zhu, Z. R., Collins, L. B., Wyrwoll, K. H. & Eichstätter, R. The Last Interglacial sea level change: new evidence from the Abrolhos islands, West Australia. *Geol. Rundschau* **85**, 606–614 (1996).
64. O’Leary, M. J., Hearty, P. J. & McCulloch, M. T. Geomorphic evidence of major sea-level fluctuations during marine isotope substage-5e, Cape Cuvier, Western Australia. *Geomorphology* **102**, 595–602 (2008).
65. Stirling, C. H., Esat, T. M., Lambeck, K. & McCulloch, M. T. Timing and duration of the Last Interglacial: evidence for a restricted interval of widespread coral reef growth. *Earth Planet. Sci. Lett.* **160**, 745–762 (1998).
66. Stirling, C. H., Esat, T. M., McCulloch, M. T. & Lambeck, K. High-precision U-series dating of corals from Western Australia and implications for the timing and duration of the Last Interglacial. *Earth Planet. Sci. Lett.* **135**, 115–130 (1995).
67. Zhu, Z. R. *et al.* High-precision U-series dating of Last Interglacial events by mass spectrometry: Houtman Abrolhos Islands, western Australia. *Earth Planet. Sci. Lett.* **118**, 281–293 (1993).
68. Whitney, B. B. & Hengesh, J. V. Geomorphological evidence of neotectonic deformation in the Carnarvon Basin, Western Australia. *Geomorphology* **228**, 579–596 (2015).
69. Hay, C. C. *et al.* The sea-level fingerprints of ice-sheet collapse during interglacial periods. *Quat. Sci. Rev.* **87**, 60–69 (2014).
70. Austermann, J., Mitrovica, J. X., Latychev, K. & Milne, G. A. Barbados-based estimate of ice volume at Last Glacial Maximum affected by subducted plate. *Nat. Geosci.* **6**, 553–557 (2013).
71. Montaggioni, L. F. & Hoang, C. T. The last interglacial high sea level in the granitic Seychelles, Indian ocean. *Palaeogeogr. Palaeoclimatol. Palaeoecol.* **64**, 79–91 (1988).
72. Israelson, C. & Wohlfarth, B. Timing of the Last-Interglacial high sea level on the Seychelles Islands, Indian Ocean. *Quat. Res.* **51**, 306–316 (1999).
73. Edinger, E. N., Burr, G. S., Pandolfi, J. M. & Ortiz, J. C. Age accuracy and resolution of Quaternary corals used as proxies for sea level. *Earth Planet. Sci. Lett.* **253**, 37–49 (2007).
74. Kidwell, S. M. Time-averaging and fidelity of modern death assemblages: building a taphonomic foundation for conservation palaeobiology. *Palaeontology* **56**, 487–522 (2013).
75. Hopley, D., Smithers, S. G. & Parnell, K. *The Geomorphology of the Great Barrier Reef: Development, Diversity and Change.* (Cambridge University Press, 2007).
76. Camoin, G. F. & Webster, J. M. Coral reef response to Quaternary sea-level and environmental changes: State of the science. *Sedimentology* **62**, 401–428 (2015).
77. Woodroffe, C. D. & Webster, J. M. Coral reefs and sea-level change. *Mar. Geol.* **352**, 248–267 (2014).
78. Braithwaite, C. J. R. *et al.* Origins and development of Holocene coral reefs: a revisited model based on reef boreholes in the Seychelles, Indian Ocean. *Int. J. Earth Sci.* **89**, 431–445 (2000).
79. Kerans, C., Zahm, C., Bachtel, S. L., Hearty, P. & Cheng, H. Anatomy of a late Quaternary carbonate island: Constraints on timing and magnitude of sea-level fluctuations, West Caicos, Turks and Caicos Islands, BWI. *Quat. Sci. Rev.* **205**, 193–223 (2019).
80. Plaziat, J.-C., Reyss, J.-L., Choukri, A. & Cazala, C. Diagenetic rejuvenation of raised coral reefs and precision of dating. The contribution of the Red Sea reefs to the question of reliability of the Uranium-series datings of middle to late Pleistocene key reef-terraces of the world. *Carnets Géologie / Notebooks Geol.* **4**, 2008/04 (2008).
81. Plaziat, J.-C. *et al.* Quaternary changes in the Egyptian shoreline of the northwestern Red Sea and the Gulf of Suez. *Quat. Int.* **29/30**, 11–22 (1995).
82. Plaziat, J.-C. *et al.* Mise en évidence, sur la cote récifale d’Égypte, d’une regression interrompant brièvement le plus haut niveau du dernier interglaciaire (5e); un nouvel indice de variations glacio-eustatiques a haute fréquence au Pleistocene? *Bull. la Société Géologique Fr.* **169**, 115–125 (1998).

83. Bruggemann, H. J. *et al.* Stratigraphy, palaeoenvironments and model for the deposition of the Abdur Reef Limestone: context for an important archaeological site from the last interglacial on the Red Sea coast of Eritrea. *Palaeogeogr. Palaeoclimatol. Palaeoecol.* **203**, 179–206 (2004).
84. Walter, R. C. *et al.* Early human occupation of the Red Sea coast of Eritrea during the last interglacial. *Nature* **405**, 65–69 (2000).
85. Cutler, K. B. *et al.* Rapid sea-level fall and deep-ocean temperature change since the last interglacial period. *Earth Planet. Sci. Lett.* **206**, 253–271 (2003).
86. Hibbert, F. D. *et al.* Coral indicators of past sea-level change: A global repository of U-series dated benchmarks. *Quat. Sci. Rev.* **145**, 1–56 (2016).
87. Siddall, M., Smeed, D. A., Matthiesen, S. & Rohling, E. J. Modelling the seasonal cycle of the exchange flow in Bab El Mandab (Red Sea). *Deep. Res. Part I Oceanogr. Res. Pap.* **49**, 1551–1569 (2002).
88. Bates, S. L., Siddall, M. & Waelbroeck, C. Hydrographic variations in deep ocean temperature over the mid-Pleistocene transition. 2002–2004 (2011).

Reviewers' comments:

Reviewer #2 (Remarks to the Author):

Response to the response of Rohling et al: Asynchronous Antarctic and Greenland ice-volume contributions to the LIG SL highstand

First, my comment stated that the Authors claimed, and continue to claim that:

"The intensively studied, sampled, and dated LIG coral/reef records of the Seychelles 8–10, Bahamas 15,19, and Western Australia 18,69–73 give an emerging picture of LIG sea level. These records are especially useful given that: (1) they span extended periods of the LIG, (2) they have relatively high temporal sampling and density of radiometric dating, (3) they are from tectonically stable areas; (4) they have well documented stratigraphic superposition of LIG units, and (5) for the Seychelles, there are well-constrained palaeo-water depth estimates." [Synthesis SI]

Their analysis uncritically accepts the claims of these studies yet critically dismiss other models of LIG sea level that don't fit with their RSR data.

"..We did not use the well-dated and stratigraphically well-characterised Yucatan record (Blanchon et al., 2009ref.62) because this is a backstepping reef sequence (i.e., no superposition of LIG reef units, and therefore no strict stratigraphic control), and because the authors did not provide any information on the palaeo-water depth of the corals dated."

Yet they fail to cite the most detailed study of the Yucatan site (Blanchon 2010 Coral Reefs 29:481–498, DOI 10.1007/s00338-010-0599-0) where continuity between reef units is clearly demonstrated proving a strict stratigraphic control, and where the depth limitations of the corals is provided along with elevations of the reef crest, (which provide accurate SL positions of stillstand during the LIG). So, at the risk of repeating my original comment, these authors have cherry picked the studies they want to represent LIG SL history (Bahamas, Seychelles) and use them to confirm their RSR findings. This is confirmation bias: an uncritical acceptance of the claims made in papers supporting their argument and a dismissal of arguments and models that do not fit their data.

Furthermore, in their response, the authors now state that their minds are made up and that: "... It is quite clear from those representations that there is no one single, perfect LIG sea-level site, as we now specifically highlight in lines 87-90), but that there is an emerging pattern." Obviously this 'emerging pattern' is based on their uncritical acceptance of the papers that fit, and dismissal of papers that don't. The fact of the matter is that there are at least 2 well-documented models of LIG SL based on reef data. So instead of picking sides why not just acknowledge this in the body of the paper and provide a balanced assessment of the published literature on this topic (and not just paraphrase the original articles as has been done in the SI).

Second, the Authors then address my comment on their uncritical acceptance of studies that failed to consider stratigraphic consistency of ages and rates of framework development. They respond that:

"...There are no straightforward stratigraphic relationships between individual coral colonies.... coral reefs grow in 3-dimensions, so that a prograding reef may contain a wide range of ages along the same elevation (or terrace), and these may be interlocked in a complex 3-dimensional manner. The implication is that data cannot be rejected where multiple ages are observed for a single elevation (as opposed to for a single coral sample). Likewise, age inversions that are identified on the basis of age and elevation data alone cannot be rejected out of hand because such relationships may be consistent with 3-dimensional development of the reef." No references are provided for their 'chaotic model' of prograding reef development. They continue that:

"...With respect to the comment that coral ages should vary by less than 1,000 years per unit, we emphasize that this value was obtained for a specific reef in one specific location (Huon Peninsula, Papua New Guinea), where the total age range per horizon was found to be 800–1060 year (Edinger et al., 2007ref.73)."

So in other words, they dismiss published work as 'just one study, on one reef' and prefer their

apparently imaginary 'chaotic model' where corals of any age can live side by side in the same horizon. One is left with the impression that these authors will go to any lengths to dismiss valid scientific work that provides inconvenient data.

The rest of their response is a lengthy but unconvincing argument as to why none of the review comments I made apply to their use of reef data to support their claims. And to be honest, I have neither the time nor inclination to address them, other than to say they follow the same *modus operandi* as alluded to above.

If the editor wants to accept this paper based on the other reviews, my suggestion would be to require that the authors state in the body of the text that the Yucatan LIG reef sequence is a valid and well-supported competing model of LIG sea level but is at odds with their preferred scenario (ie, a double highstand punctuated by a ephemeral SL fall). In other words, provide a balance view of the LIG reef literature, not the biased view they provide.

Reviewer #3 (Remarks to the Author):

The revised document is an improvement on the original. The new manuscript, combined with the response to the reviews, addresses the issues raised.

Reviewer #4 (Remarks to the Author):

Summary

Overall, I think the authors have responded well to the reviewers comments. For a number of points, I do have some concerns, for which I give additional comments to the first review and provide some additional comments. I think it is a well written manuscript, but I do think that several points need to be clarified or more critically discussed in the manuscript before it can be published. Specifically, the points raised by reviewer 1 under L171, L192, and my two main points below.

Regarding the rebuttal, answers to reviewer 1.

L23: The authors have not responded to the reviewer's comment on NH ice sheets contributions after 130 kyr and before 115 kyr BP. This is slightly mentioned in the caption of Figure 3, but I feel also that they can contribute to variability above -10 m (L393). This should be included in the text when discussing the residual curve of Figure 3b.

L90: Yes, well responded. See for some additional comments from my part below.

L111: Good response. Again, I do miss the discussion of other effects influencing the GMSL-GrIS signal. This needs to be addressed, as I mention below.

L129: Addressed well.

L137: Addressed well.

L141: Addressed well.

L171: In general, I think your response focusses on discussing evidence for WAIS retreat during the LIG than on answering the question of reviewer 1. In the discussion of the palaeoceanographic data, I think you should include a bit more how this possible link between meltwater release and the MAR of site 1094 can be established and/or masked, not only the later lack of variability in MAR (as pointed out by Reviewer 1), but also the high peak during the deglaciation, prior to HS11 as shown in Figure 4g. Also, most data you refer to in Figure 4 do show that there is clear

interglacial signal around Antarctica, but I do not see a clear linkage of this data to the high amplitude variability. This should be mentioned in the text (from Line 200 onwards).

L181: Addressed well.

L189: Addressed well.

L192: I agree with Reviewer 1 and would request serious alteration to the text, both in the main text and supplement. In terms of the answers given by the author, I don't feel you fully answer the reviewers comments. Specifically, your point 1 the GIA feedback actually causes your retreat phase to be slower, so further away from the large rates shown by the RSL record. Even if the WAIS was completely removed, large parts of the AIS will still considerable produce icebergs.

As reviewer 1 also argued, current research of the AIS, including strong physics for ice sheet retreat, do not show this large variability on such relatively short time scales. If it would be fully the AIS, the curves in Figure 3b go from a really warm Pliocene AIS (> 10 m sle) to an almost LGM configuration (closing in on - 10 m sle) within 7000 years! Also, the discussion under #6 in the supplement, you argue that "mass loss would need to become almost zero". I think this statement should be completely removed! As Reviewer 1 states, this is an unrealistic case. If the GrIS or AIS are to be smaller than present, this is due to warming and thus an increase in mass loss. On the contrary, mass loss through iceberg calving will actually increase when the ice sheets are advancing.

For your ball-park assessment, if you consider the state of today's ice sheets, I think most importantly you need to consider the total mass budget of the ice sheets and cannot consider mass gain (through snow accumulation) alone. Hence in the current climate, the maximum contribution to sea level from the AIS was 0.73 ± 0.31 mm/year, and GrIS was ~ 1.2 mm/yr during the past decade (WCRP, 2018).

If you would only consider the mass gain. For Antarctica (WCRP, 2018) this is about 2200 Gt/yr, which is over -0.6 m/c, the GrIS gains about 600 Gt/yr (Van den Broeke, 2016), which is ~ -0.16 m/c (but again, surface melt is considerable!). The numbers you mention in SI Part 6 (0.15 m/y for AIS and 0.30 m/y for GrIS) are not correct.

Similar to the reviewers statement, I would like to see a statement (or similar to) as reviewer 1 suggests: ".. and acknowledge that there exists a real problem to reconcile the large sea-level variations in your records with current understanding of ice sheet physics."

The sentence in Lines 269-272 should be revised to something like:
"Given that the sea-level signal is a net result of ice build-up and ice loss, considerable rates of sea-level lowering are hard to reconcile due to expected increase in iceberg calving when the ice sheet is advancing".

Reference

WCRP Global Sea Level Budget Group, 2018. Global sea-level budget 1993--present, Earth System Science Data, 10, 1551-1590. doi: 10.5194/essd-10-1551-2018

L199: Addressed well.

L301: Addressed well.

Review

Below are a number of short comments to be addressed by the authors.

Line 40: Please define 'a little'.

Line 45-47: You also use numerical modelling results (for the GrIS) so this statement should be adjusted. I do think that it is required to include numerical models to fully disentangle the contributions from the AIS and GrIS, either climate, ice sheet or GIA models, or a coupled model including these components.

Lines 178: To me the amplitude, especially the peaks between 130 and 127 ka do not 'seem' larger but definitely are larger. This should be reworded in the text.

Line 287-288: I wouldn't say the agreement with the high, relative short-term, variability is supported by the palaeoceanographic data.

Main comments

Besides the comments raised by reviewer 1, I also have two main concerns with the discussion of the RSL variability. I hope the author can address these remarks in the manuscript and provide a more critical discussion on their interpretation of the results.

1) On the absolute numbers and large variability

After reading the paper, I do feel your conclusions are largely based on two things: 1) the variability of the RSL itself, which is not supported (yet) by other sites, as you discuss. And 2) the GrIS contribution used to derive the AIS variability shown in Figure 3.

In terms of the variability, the absolute numbers are not discussed. I do think it is a very nice RSL record, and your statistical methods seem robust. But I do question of these high temporal and amplitude variability is a real signal of sea level caused by the AIS alone. Do you think they are a real sea level signal, only caused from ice sheets? I would plea that they are a result of regional variability that cannot be ascribed to the AIS alone. As Reviewer 1 pointed out a couple of times, the large variability is hard to reconcile with current understanding of ice sheet dynamics.

Currently the discussion (and in the Conclusions, paragraph starting at Line 301), largely focusses on possible causes in the polar regions, whereas I think that there lies a more regional cause in the high amplitude, and millennial scale variability. Therefore, I think in the discussion you should mention the robustness of the absolute numbers: Is it an amplified signal, resulting from perhaps ocean dynamics or local runoff or something influencing the original d18O data? Specifically, I think you cannot name the curve in Figure 3b 'AIS' because to me it is clear that the variability that is shown by the green/orange curves is not AIS alone.

2) The choice of the GrIS contribution

For the GrIS contribution, I would like to see some more discussion. There are quite a number of studies that derive a (time varying) GrIS contribution during the LIG. These should be mentioned (referred to) in the introduction, around lines 43-45. As reviewer 1 also pointed out, I don't think the GrIS will differ very much, and most modelling studies more or less agree on the GrIS contribution during the LIG. But contributions are not constant initially I think, especially the studies that include longer simulations (Tabone, Goelzer).

From line 128 onwards, a wider discussion could be added on the GrIS contribution. Below a number of the most recent examples. I do not require you to use all of the studies listed below.

LIG Greenland modelling estimates:

- Stone, E. J. et al., 2013. Quantification of the Greenland ice sheet contribution to Last Interglacial sea level rise, *Climate of the Past*, 9, 621-639. doi: 10.5194/cp-9-621-2013
- Helsen, M. M. et al., 2013. Coupled regional climate--ice-sheet simulation shows limited Greenland ice loss during the Eemian, *Climate of the Past*, 9, 1773-1788. doi: 10.5194/cp-9-1773-2013
- Calov, R. et al., 2015. Simulating the Greenland ice sheet under present-day and palaeo constraints including a new discharge parameterization, *The Cryosphere*, 9, 179-196. doi:

10.5194/tc-9-179-2015

- Goelzer, H. et al., 2016. Last Interglacial climate and sea-level evolution from a coupled ice sheet--climate model, *Climate of the Past*, 12, 2195-2213. doi: 10.5194/cp-12-2195-2016

- Tabone, I. et al., 2018. The sensitivity of the Greenland Ice Sheet to glacial--interglacial oceanic forcing, *Climate of the Past*, 14, 455-472. doi: 10.5194/cp-14-455-2018

- Bradley, S. L. et al., 2018. Simulation of the Greenland Ice Sheet over two glacial--interglacial cycles: investigating a sub-ice-shelf melt parameterization and relative sea level forcing in an ice-sheet--ice-shelf model, *Climate of the Past*, 14, 619-635. doi: 10.5194/cp-14-619-2018

Response to reviewers

Rohling et al., "Asynchronous Antarctic and Greenland ice-volume contributions to the last interglacial sea-level highstand" *N. Comms.* (round 2)

We thank the reviewers for their thorough and useful comments. Our responses to specific comments are given below in blue, with actions highlighted in yellow.

REVIEWER 2

We thank Reviewer 2 for his/her comments.

Our paper primarily is about the asynchronous contribution of the polar ice sheets to LIG sea level, *not* about how well the discrete coral data compare with each other and with the Red Sea record. If we were to remove comparison with the coral data entirely, then our conclusions would still stand. However, ignoring the wealth of hard-won coral data would constitute a serious disservice to the wider palaeo community. Moreover, any reader would immediately (and reasonably) ask why we did not evaluate the coral-based information. This is why we present an integration of: (i) our Red Sea records; (ii) the (often divergent) coral and other evidence of LIG sea-level instability (main text and *Supplementary Information*); (iii) palaeoceanographic evidence from both hemispheres that indicates polar ice-sheet instability; (iv) brief discussion of ice-sheet mass balance and mechanisms of change/feedbacks that may go some way to explain such sea-level oscillations. Inclusion of all these aspects in the manuscript is essential, given that they are all interconnected, and that any inferred LIG sea-level variability eventually needs to satisfy *all* of these aspects (or usefully explain why some are not satisfied). Thus, we consider it unfortunate that Reviewer 2 only focusses on the coral aspects of the study, ignoring our effort to integrate the various strands of evidence.

We were disappointed to see the accusations of "*confirmation bias*", "*uncritical acceptance of papers*", "*cherry-picking*" and that we would "*go to any lengths to dismiss valid scientific work that provides inconvenient data*". In fact, we explicitly acknowledge (in several places, lines 80-83, 97-99 and *Supplementary Information*, Part 1) that the occurrence, magnitude and timing of and LIG sea-level oscillations is far from settled in the coral data, because of divergent and often contradictory field evidence. We have endeavoured to provide an overview in the supplement (covering 80+ papers in this section alone), which is more extensive than we have found anywhere else. In review round 1, both Reviewers 1 and 3 found the *Supplementary Information* extensive and detailed, which is diametrically opposite to the statements of Reviewer 2. Given that the coral sea-level field is moving toward accepted data-quality criteria, we clearly stated those underpinning our selection coral records. We were even more stringent than many recent studies in that we required not only U-series data quality and density, and elevation-criteria, but also a criterion of clearly described stratigraphic context. Reviewer 2 argues that the Yucatan data series (Blanchon 2010^{ref.1}) should have been included. We were reluctant about this because in Blanchon et al. (2009)^{ref.2} it is described as a backstepping sequence, not a stratigraphically continuous one.

Regardless, we thank the reviewer for bringing the Blanchon (2010)^{ref.1} paper back to our attention, and for completeness we have added this to the *Supplementary Information*. Given Reviewer 2's comments that we "*take sides*", we understand that we had still insufficiently emphasised the divergence in coral data for the LIG after the first revision. The relevant specialists will need to reconcile those divergences, and as suggested by the reviewer, all we can do is further emphasize the debate within the coral community. We have therefore removed the coral compilation graph from main text Figure 2. Alternate models are now listed in the *Supplementary Information* section 1 synthesis, as: (i) relatively stable sea level (i.e., one major peak) (e.g., Stirling et al., 1998^{ref.3}); (ii) two peaks separated by a sea level fall of various magnitudes (e.g., Chen et al., 1991^{ref.4}, Stein et al., 1993^{ref.5}, Sherman et al., 1993^{ref.6}, Plaziat et al., 1998^{ref.7}, Bruggemann et al., 2004^{ref.8}, Thompson and Goldstein 2005^{ref.9}, Hearty et al 2007^{ref.10}, Kerans et al., 2019^{ref.11}); (iii) relatively stable (possibly with a small drop) sea level with a rapid late rise (e.g., Neumann and Hearty, 1996^{ref.12},

Hearty, 2002^{ref.13}, O'Leary et al., 2013¹⁴, Blanchon et al., 2009^{ref.2}) and; (iv) multiple peaks (e.g., Thompson et al., 2011^{ref.15}, Rohling et al., 2008^{ref.16}, this work).

We were remiss to not provide sufficient references for our what Reviewer 2 terms our "chaotic" model of reef development. We thank the reviewer for pointing this out, and rectify this omission in the revised *Supplementary Information*. This section sought to highlight for the non-specialist community (given the multiple facets of the manuscript) the difference in depositional regimes for coral reefs and sediment cores, where the former does not necessarily accumulate monotonically, and most interpretation is predicated on the assumptions that fossil reefs are autochthonous, and that fossil assemblages faithfully record living assemblages (e.g., Pandolfi et al., 1995^{ref.17}, Greenstein and Pandolfi, 2003^{ref.18}). We hope that this addition to the *Supplementary Information* clears up this misunderstanding.

As for the our "dismissal" of the Erdinger et al. (2007)^{ref.19} study of reef accretion, as "just one study, on one reef". Indeed, we do highlight that this is one study from a particular location. We felt compelled to make that point in light of the complex, and often interacting factors that govern coral/reef growth (e.g., Dullo, 2005^{ref.20} and discussion above).

However, looking at rates of reef accretion was a good suggestion and so, rather than dismissing the comment (as is implied in the second review), we went on to demonstrate that LIG accretion rates for the Seychelles are consistent with Holocene accretion rates for that same location, and also in line with Holocene accretion rates for (relatively) nearby sites in Mauritius and the Seychelles (0.031 cm/yr; Camoin and Webster, 2015^{ref.21}). Of course, our (and any other) calculation of reef accretion rates is subject to the following important caveats: (i) that reworking and/or displacement has not significantly disturbed the reef (e.g., Davies, 1983^{ref.22}), and (ii) that there are minimal coring artefacts (e.g., Blanchon and Blakeway, 2003^{ref.23}).

In summary, we appreciate Reviewer 2's expert insights, though we object to their suggestions that we are being dishonest. The majority of criticism/concerns that were raised in the first review ought to be settled with the authors of the original studies. Regardless, we attempted to address those concerns in our first response, only to be dismissed with "the rest of their response is a lengthy but unconvincing argument... [unsubstantiated opinion with no references] ...and to be honest, I have neither the time nor the inclination to address them, other than to say that they follow the same *modus operandi* [dismissal of inconvenient data?] alluded to above". We feel that this is (1) a bit strong and misrepresenting our effort, (2) at odds with the other reviewers, (3) based on an assessment of only a fraction of our study.

Still, the actual comments were helpful, and we have now accordingly:

- (a) brought the omitted study into our synthesis,
- (b) transparently presented the other LIG sea-level hypotheses from coral data, and
- (c) provided a richness of references about why we state that reefs have more complicated stratigraphic make-up than deep-sea sediment cores from hardly bioturbated basins.

We hope that this alleviates the reviewer's concerns.

Reviewer 3

Thank you. The revised manuscript ("*an improvement on the original*") benefited from your useful and insightful comments.

Reviewer 4

We thank the reviewer for their insightful comments and the opinion that, in our first response, we "*responded well to the reviewers comments*".

In response to the concerns regarding our initial response (to Reviewer 1), our comments are as follows:

L23: NH ice sheets contributions after 130 kyr and before 115 kyr BP: We understand this to mean Northern Hemisphere glaciation toward the end of the LIG. Where we talk about sea level below -10 m, there are indeed issues to contend with regarding re-glaciation in the North, as was explained in the caption of Figure 3. We fully agree that this should have been discussed in the text, and have accordingly brought that into lines 175-181.

L90: Thank you. We have addressed your additional comments below.

L111: Thank you. Our comments to your queries regarding effects influencing the GMSL-GrIS signal are given below. See response to Line 23.

L129, 137 & 141: Thank you.

L171: Southern Ocean authigenic uranium (aU) mass accumulation rate (MAR) changes – reasons for the lack of late LIG changes, if this is an Antarctic meltwater signal: Our response focused on the West Antarctic Ice Sheet (WAIS) due to the geographic proximity of the core site (IODP Site 1094). If this is a meltwater signal (and reasons for this interpretation are given below), then the most likely source would be the WAIS. Sadly, only a handful of palaeoceanographic records exists from locations proximal to the various Antarctic Ice Sheet (AIS) sectors (e.g., Wilson et al., 2018^{ref.24}, McKay et al., 2012^{ref.25}) (for reasons such as logistical difficulties of recovering cores from these regions, poor preservation and/or low sedimentation rates, and the difficulty of dating these sediments; NB. we are in a major logistics proposal at the moment to remedy this omission in the existing core availability). Future development of more records is essential for revealing provenance of the meltwater pulses, and of their Antarctic-wide nature. This is not something anybody can answer yet.

Basis for interpreting the IODP Site 1094 aU MAR signal as meltwater release and how this might be “masked”: For background, though this is covered in Hayes et al. and we do not repeat it in our manuscript. The interpretation of the uranium MAR as a meltwater signal was made as authigenic uranium (aU), which is produced at or below the sediment-water interface, is a sensitive proxy for the oxygen content of sediment pore waters (e.g., Klinkhammer and Palmer, 1991^{ref.26}). Therefore changes in the aU MAR can be used as a ‘tracer’ of changes in the ventilation of the oceans, especially where sediment flux (potential dilution of aU) and the supply of organic matter (which enhances aU deposition) can be controlled for (as was done by the original authors, Hayes et al., 2014 by calculating a ‘focusing factor’ from $^{230}\text{Th}_{\text{xs}}$ and the time integrated production of ^{230}Th , and ^{230}Th normalised opal flux, respectively). This indicates that, during the interglacial, the spike in aU MAR seen in IODP 1094 cannot be explained by dilution, nor by enhanced aU production with increased organic matter (vertical and lateral supplies of organic matter decreased during this interval, Hayes et al., 2014^{ref.27}). Therefore, this spike is attributed to decreased bottom water oxygen content as a consequence of reduced Antarctic Bottom Water (AABW) ventilation in response to freshening of Antarctic coastal waters (Hayes et al., 2014^{ref.27}, Fogwill et al., 2014^{ref.28}) (and indications for this freshening and decreased AABW ventilation associated with increase surface-deep water stratification come from planktonic $\delta^{18}\text{O}$ and planktonic-benthic $\delta^{18}\text{O}$ contrasts, as shown in our Figure 4).

Lack of variability during the LIG: To explain in more detail: A lack of variability within the later LIG portion of the IODP Site 1094 aU MAR record could result from one, or a combination of the following:

- (i) loss or significant reduction of WAIS, or other sectors of the AIS, so that there is little or no suppression of AABW formation from meltwater (the sortable silt measured in the Agulhas basin^{27,29}, which closely mirrors the trend in aU MAR, as well as the similar increases in $\delta^{13}\text{C}$ IODP Sites- 1094²⁷ (South Atlantic), -1089³⁰ (Agulhas basin), -1063³¹ (Bermuda Rise) and, MD03-2664³² (Eirik Drift, North Atlantic) (our figure 4i) supports reduced freshwater inputs and a return to more vigorous AAWB (and NADW) formation);
- (ii) dilution of the aU signal from sediment transport (however, the focusing factor remains low throughout the 118 to 125 ka interval, Hayes et al., 2014^{ref.27}) and;

- (iii) reduced organic flux suppressing the formation of aU (the ^{230}Th normalised opal flux continues throughout the LIG, although at lower values than for the peak in aU MAR at 125 ka). Given the supporting $\delta^{13}\text{C}$ and sortable silt data, a reduced amount of meltwater input (and hence resumption/increased of AADW formation and therefore increased bottom water oxygenation) seems the more likely.

We have clarified this discussion in the manuscript (lines 226 to 249 and lines 256 to 268) to make the link between the aU MAR and other palaeoceanographic data presented more explicit.

High aU MAR peak during the penultimate deglaciation: This is also covered in Hayes et al. (hence, not for us to repeat). To explain in more detail: During the interval prior to HS11, the aU MAR increases due to the poor bottom water oxygenation of the AADW (as there is no enhanced supply of organic material, i.e., low opal flux²⁷, and aU production is not enhanced). The low oxygen content (i.e., poor ventilation of the bottom waters) could arise from other processes (other than meltwater) that cause enhanced stratification of the ocean such as thick and year-round sea-ice cover, or wind driven changes in upwelling over the Antarctic Zone of the Southern Ocean, as have been demonstrated for that last glacial and deglacial intervals e.g.,^{28,33–36}. The increased aU MAR during the deglacial coincided with with an increase in the supply of organic material (increased opal flux) which also enhanced aU production at the site (Hayes et al., 2014). The subsequent decline in this first peak in aU MAR was driven by the increased oxygenation of the bottom waters as overturning increased (i.e., increased ventilation) as the deglaciation progressed²⁷.

It goes beyond the scope of our study to fully elaborate on the aU MAR record, especially given that it would be a repeat of what is in the Hayes et al (2014) study. As stated above, we have clarified aspects of this discussion in the manuscript (lines 226 to 249 and lines 256 to 268) to make the link between the aU MAR and other palaeoceanographic data presented more explicit.

L181 & 189: Thank you.

L192: High rates of sea-level rise, and large variability on relatively short timescales: This comment (from both Reviewer 1 and 4) stems from the difficulty, at present, to reconcile both the magnitude and rates of sea-level change documented in the palaeo record for the Last Interglacial (from separate locations, e.g., our study, Blanchon et al., 2009^{ref.2}, O'Leary et al 2013^{ref.14}, Dutton et al., 2015^{ref.37}), and derived from statistical modelling of multiple sites (e.g., Kopp et al 200^{ref.38}, 2013^{ref.39}, Düsterhus et al., 2016^{ref.40}) with currently existing ice sheet models/ice physics.

This can no longer be explained away in terms of uncertainties in just the data, given that high variability is seen in many different sources and approaches. Therefore, the disagreement highlights that:

- (a) changes in the Last Interglacial were unprecedented relative to the modern/observational era (but may possibly be approached in the future);
- (b) the record of modern direct observations of ice sheet mass loss are simply of insufficient length to disentangle the various ice sheet forcing mechanisms and feedbacks (e.g., Steig and Neff, 2018^{ref.41}, Bamber et al., 2019^{ref.42}), especially as ice sheets have a long response time, and are severely lagging behind equilibrium with present climate trends and;
- (c) there remain uncertainties (especially unknown unknowns) in ice dynamics (e.g., Bamber et al., 2019^{ref.42}) and/or additional, as yet poorly understood processes that need to be incorporated into models (e.g., hydrofracturing, ice cliff failure, DeConto and Pollard, 2016; rifts below the ice shelves that enable the penetration of warm water, and hence erosion of the ice shelf base, Alley et al., 2016^{ref.43}, Jeong et al., 2016^{ref.44}; solid Earth and gravitational processes, e.g., Gomez et al., 2015^{ref.45}, Konrad et al., 2015^{ref.46}).

The latter (c) may be especially important when identifying the high-end and/or rapid mass loss in past warm intervals (e.g., the collapse of the reverse bed slope marine sectors of the AIS^{e.g., refs.47,48}). These factors have contributed to the continuing uncertainty surrounding the contribution of ice sheets to past (and future) sea level rise (e.g., IPCC AR5^{ref.49}, Bamber and Aspinall, 2013^{ref.50}). We also note with interest

the drastic increase in the range (total magnitude) of potential ice-sheet contributions estimated for future sea-level rise, given recent attempts to refine process understanding and representation within ice sheet models (Bamber et al., 2019⁴²). This illustrates how observations cannot be dismissed, but need to be used to identify discrepancies with modelling results, so that we can try to work out how reconciliation might be achieved. The palaeo record puts strong bounds on this ongoing debate because it (and it alone) incorporates all known and unknown processes and feedbacks, including those that act over a variety of timescales (i.e., equilibrium response of the ice sheets to climate forcing).

Hence, we stress the importance of getting strong data constraints out into the public domain, so they can be used to challenge and interrogate models. This process is going to be very important for working out potential future sea-level change. **Regardless, we have changed the main text (lines 274 to 292) and Supplementary Information section 6 to be more in line with the reviewer's suggestions and references – with thanks.**

In response to specific queries:

(i) GIA feedback: GIA feedback slowing mass loss: Reviewer 4 seems to be confused. We do not argue that this mechanism helped cause the high rates of rise. Instead, we argue that it provided a negative feedback that had stabilising effect on grounding line retreat (e.g., Gomez et al., 2010^{ref.51}). The negative feedback is inferred to explain the dual rise structure (hence, it causes the “interruption” of the rise). This mechanism modulated AIS retreat during the Holocene (Kingslake et al., 2018)^{ref.52}, and is important when considering future evolution of the AIS, especially the marine based WAIS. However, **most studies that predict unstable ice mass loss do not include GIA feedbacks** (i.e., that help to prevent runaway loss from unstable configurations) because this is computationally non-trivial and complicated by coupling difficulties (e.g., de Boer et al., 2017)⁵³. We emphasize this mechanism because it acts on timescales that are important for estimating sea-level contributions to the LIG, and to provide impetus for including this mechanism in LIG (and future) modelling. **We have retained mention to this, but clarified the context to make this clearer (lines 274 to 277).**

(ii) Even if the WAIS was completely removed, large parts of the AIS will still considerable produce icebergs: Agreed, mass loss from AIS sectors other than the WAIS would be possible. However, there is currently a paucity of observational data (particularly proximal sediment cores) with which to test this. To be complete, all that exists now is (though this goes beyond the scope of the manuscript):

- *WAIS empirical evidence:* McKay et al., 2012^{ref.25}: suggestion that the Ross ice sheet was lost sometime during the last 240 ka, potentially during the LIG. NB. in some modelling studies, the loss of the Ross ice shelf results in significant loss from the WAIS (e.g., DeConto and Pollard, 2016^{ref.54})
- *EAIS empirical evidence:* Nd in sediments off East Antarctica (IODP Site U1361), suggests significant thinning or margin retreat into the Wilkes subglacial basin during the LIG (Wilson et al., 2018^{ref.24}).

(iv) Strong physics for ice sheet retreat: See our response above under Line 192.

(v) Mass balance calculations: **We have changed the main text (lines 277 to 281) and Supplementary Information section 6 to be more in line with the reviewer's suggestions and references – with thanks. We have also started Supplementary Information Section 6 with the requested sentence that it is difficult to reconcile sea-level variations and model physics, as requested by Reviewers 1 and 4.**

L199 & L301: Thank you.

Other comments from Reviewer 4 (in addition to those raised by Reviewer 1)

L40: definition of a "little". Thanks, we have clarified this in the text to "up to 2 m".

L45 to 47: As suggested by the reviewer, we have amended the text to include mention of numerical modelling (climate, ice sheet, GIA or some combination of all of these)

L178: Thanks – corrected.

L287-288: agreement (or not) of the "high, relative short-term variability" in the Red Sea sea-level record with the palaeoceanographic data. We have reworded this to clarify that we refer here to Hayes et al. in particular.

Main comment 1: Absolute numbers and large variability of sea level change: See response above under "Line 192".

Main comment 2: choice of Greenland Ice Sheet component

We find that the Yau et al. paper is a reasonable match, within uncertainties, for other modelling studies (some of which we have now included in the general citations of the manuscript), and especially the $\delta^{18}\text{O}_{\text{sw}}$ observations are very supportive of the structure as well. We agree that smaller Greenland sea-level contributions have been suggested in the other studies than by Yau et al., but note that our $\delta^{18}\text{O}_{\text{sw}}$ reconstruction similarly falls lower (but within combined uncertainties). We emphasise this in lines 121 to 124.

References cited:

1. Blanchon, P. Reef demise and back-stepping during the last interglacial, northeast Yucatan. *Coral Reefs* **29**, 481–498 (2010).
2. Blanchon, P., Eisenhauer, A., Fietzke, J. & Liebtrau, V. Rapid sea-level rise and reef back-stepping at the close of the last interglacial highstand. *Nature* **458**, 881–884 (2009).
3. Stirling, C. H., Esat, T. M., Lambeck, K. & McCulloch, M. T. Timing and duration of the Last Interglacial: evidence for a restricted interval of widespread coral reef growth. *Earth Planet. Sci. Lett.* **160**, 745–762 (1998).
4. Chen, J. H., Curran, H. A., White, B. & Wasserburg, G. J. Precise chronology of the last interglacial period: ^{234}U - ^{230}Th data from fossil coral reefs in the Bahamas. *Geol. Soc. Am. Bull.* **103**, 82–97 (1991).
5. Stein, M. et al. TIMS U-series dating and stable isotopes of the last interglacial event in Papua New Guinea. *Geochim. Cosmochim. Acta* **57**, 2541–2554 (1993).
6. Sherman, C. E., Glenn, C. R., Jones, A. T., Burnett, W. C. & Schwarcz, H. P. New evidence for two highstands of the sea during the last interglacial, oxygen isotope substage 5e. *Geology* **21**, 1079–1082 (1993).
7. Plaziat, J.-C. et al. Mise en évidence, sur la cote recifale d’Egypte, d’une regression interrompant brievement le plus haut niveau du dernier interglaciaire (5e); un nouvel indice de variations glacio-eustatiques a haute frequence au Pleistocene? *Bull. la Société Géologique Fr.* **169**, 115–125 (1998).
8. Bruggemann, H. J. et al. Stratigraphy, palaeoenvironments and model for the deposition of the Abdur Reef Limestone: context for an important archaeological site from the last interglacial on the Red Sea coast of Eritrea. *Palaeogeogr. Palaeoclimatol. Palaeoecol.* **203**, 179–206 (2004).
9. Thompson, W. G. & Goldstein, S. L. Open-system coral ages reveal persistent suborbital sea-level cycles. *Science* **308**, 401–404 (2005).
10. Hearty, P. J., Hollin, J. T., Neumann, A. C., O’Leary, M. J. & McCulloch, M. T. Global sea-level fluctuations during the Last Interglaciatiion (MIS 5e). *Quat. Sci. Rev.* **26**, 2090–2112 (2007).
11. Kerans, C., Zahm, C., Bachtel, S. L., Hearty, P. & Cheng, H. Anatomy of a late Quaternary carbonate island: Constraints on timing and magnitude of sea-level fluctuations, West Caicos, Turks and Caicos Islands, BWI. *Quat. Sci. Rev.* **205**, 193–223 (2019).
12. Neumann, A. C. & Hearty, P. J. Rapid sea-level changes at the close of the last interglacial (substage 5e) recorded in Bahamian island geology. *Geology* **24**, 775–778 (1996).
13. Hearty, P. J. Revision of the Late Pleistocene stratigraphy of Bermuda. *Sediment. Geol.* **153**, 1–21 (2002).
14. O’Leary, M. J. et al. Ice sheet collapse following a prolonged period of stable sea level during the last interglacial. *Nat. Geosci.* **6**, 796–800 (2013).
15. Thompson, W. G., Curran, H. A., Wilson, M. A. & White, B. Sea-level oscillations during the last interglacial highstand recorded by Bahamas corals. *Nat. Geosci.* **4**, 684–687 (2011).

16. Rohling, E. J. *et al.* High rates of sea-level rise during the last interglacial period. *Nat. Geosci.* **1**, 38–42 (2008).
17. Pandolfi, J. M. & Minchin, P. A comparison of taxonomic composition and diversity between reef coral life and death assemblages in Madang Lagoon, Papua New Guinea. *Palaeogeogr. Palaeoclimatol. Palaeoecol.* **119**, 321–341 (1995).
18. Greenstein, B. J. & Pandolfi, J. M. Taphonomic alteration of reef corals: effects of reef environment and coral growth form. II. The Great Barrier Reef. *Palaios* **18**, 495–509 (2003).
19. Edinger, E. N., Burr, G. S., Pandolfi, J. M. & Ortiz, J. C. Age accuracy and resolution of Quaternary corals used as proxies for sea level. *Earth Planet. Sci. Lett.* **253**, 37–49 (2007).
20. Dullo, W.-C. Coral growth and reef growth: a brief review. *Facies* **51**, 33–48 (2005).
21. Camoin, G. F. & Webster, J. M. Coral reef response to Quaternary sea-level and environmental changes: State of the science. *Sedimentology* **62**, 401–428 (2015).
22. Davies, P. J. Reef growth. in *Perspectives on Coral Reefs* (ed. Barnes, D. J.) 69–106 (Australian Institute of Marine Science, 1983).
23. Blanchon, P. & Blakeway, D. Are catch-up reefs an artefact of coring? *Sedimentology* **50**, 1271–1282 (2003).
24. Wilson, D. J. *et al.* Ice loss from the East Antarctic Ice Sheet during late Pleistocene interglacials. *Nature* **561**, 383–386 (2018).
25. McKay, R. *et al.* Pleistocene variability of Antarctic Ice Sheet extent in the Ross Embayment. *Quat. Sci. Rev.* **34**, 93–112 (2012).
26. Klinkhammer, G. & Palmer, M. Uranium in the oceans: Where it goes and why. *Geochim. Cosmochim. Acta* **55**, 1799–1806 (1991).
27. Hayes, C. T. *et al.* A stagnation event in the deep South Atlantic during the last interglacial period. *Science*. **346**, 1514–1517 (2014).
28. Fogwill, C. J. *et al.* Testing the sensitivity of the East Antarctic Ice Sheet to Southern Ocean dynamics: past changes and future implications. *J. Quat. Sci.* **29**, 91–98 (2014).
29. Krueger, S., Leuschner, D. C., Ehrmann, W., Schmiel, G. & Mackensen, A. North Atlantic Deep Water and Antarctic Bottom Water variability during the last 200ka recorded in an abyssal sediment core off South Africa. *Glob. Planet. Change* **80–81**, 180–189 (2012).
30. Ninnemann, U. S., Charles, C. D. & Hodell, D. A. Origin of Global Millennial Scale Climate Events: Constraints from the Southern Ocean Deep Sea Sedimentary Record. in *Geophysical Monograph Series volume 112, Mechanisms of Global Climate Change at Millennial Time Scales* (eds. Clark, P. U., Webb, R. S. & Keigwin, L. D.) 99–112 (American Geophysical Union, 1999). doi:10.1029/GM112p0099
31. Deaney, E. L., Barker, S. & van der Flierdt, T. Timing and nature of AMOC recovery across Termination 2 and magnitude of deglacial CO₂ change. *Nat. Commun.* **8**, 14595 (2017).
32. Galaasen, E. V. *et al.* Rapid Reductions in North Atlantic Deep Water During the Peak of the Last Interglacial Period. *Science*. **343**, 1129–1132 (2014).
33. Marchitto, T. M., Lehman, S. J., Ortiz, J. D., Fluckiger, J. & van Geen, A. Marine Radiocarbon Evidence for the Mechanism of Deglacial Atmospheric CO₂ Rise. *Science*. **316**, 1456–1459 (2007).
34. Anderson, R. F. *et al.* Wind-Driven Upwelling in the Southern Ocean and the Deglacial Rise in Atmospheric CO₂. *Science*. **323**, 1443–1448 (2009).
35. Skinner, L. C., Fallon, S. J., Waelbroeck, C., Michel, E. & Barker, S. Ventilation of the Deep Southern Ocean and Deglacial CO₂ Rise. *Science*. **328**, 1147–1151 (2010).
36. Fogwill, C. J. *et al.* Antarctic ice sheet discharge driven by atmosphere-ocean feedbacks at the Last Glacial Termination. *Sci. Rep.* **7**, 39979 (2017).
37. Dutton, A., Webster, J. M., Zwart, D., Lambeck, K. & Wohlfarth, B. Tropical tales of polar ice: evidence of Last Interglacial polar ice sheet retreat recorded by fossil reefs of the granitic Seychelles islands. *Quat. Sci. Rev.* **107**, 182–196 (2015).
38. Kopp, R. E., Simons, F. J., Mitrovica, J. X., Maloof, A. C. & Oppenheimer, M. Probabilistic assessment of sea level during the last interglacial stage. *Nature* **462**, 863–867 (2009).
39. Kopp, R. E., Simons, F. J., Mitrovica, J. X., Maloof, A. C. & Oppenheimer, M. A probabilistic assessment of sea level variations within the last interglacial stage. *Geophys. J. Int.* **193**, 711–716 (2013).
40. Dusterhus, A., Tamisiea, M. E. & Jevrejeva, S. Estimating the sea level highstand during the last interglacial: a probabilistic massive ensemble approach. *Geophys. J. Int.* **206**, 900–920 (2016).
41. Steig, E. J. & Neff, P. D. The prescience of paleoclimatology and the future of the Antarctic ice sheet. *Nat. Commun.* **9**, 2730 (2018).
42. Bamber, J. L., Oppenheimer, M., Kopp, R. E., Aspinall, W. P. & Cooke, R. M. Ice sheet contributions to future sea-level rise from structured expert judgment. *Proc. Natl. Acad. Sci.* 201817205 (2019). doi:10.1073/pnas.1817205116
43. Alley, K. E., Scambos, T. A., Siegfried, M. R. & Fricker, H. A. Impacts of warm water on Antarctic ice shelf stability through basal channel formation. *Nat. Geosci.* **9**, 290–293 (2016).
44. Jeong, S., Howat, I. M. & Bassis, J. N. Accelerated ice shelf rifting and retreat at Pine Island Glacier, West

- Antarctica. *Geophys. Res. Lett.* **43**, 11,720–11,725 (2016).
45. Gomez, N., Pollard, D. & Holland, D. Sea-level feedback lowers projections of future Antarctic Ice-Sheet mass loss. *Nat. Commun.* **6**, 8798 (2015).
 46. Konrad, H., Sasgen, I., Pollard, D. & Klemann, V. Potential of the solid-Earth response for limiting long-term West Antarctic Ice Sheet retreat in a warming climate. *Earth Planet. Sci. Lett.* **432**, 254–264 (2015).
 47. Ritz, C. *et al.* Potential sea-level rise from Antarctic ice-sheet instability constrained by observations. *Nature* **528**, 115–118 (2015).
 48. Mengel, M. *et al.* Future sea level rise constrained by observations and long-term commitment. *Proc. Natl. Acad. Sci. U. S. A.* **113**, 2597–2602 (2016).
 49. Church, J. A. *et al.* Sea Level Change. in *Climate Change 2013: The Physical Science Basis. Contribution of Working Group I to the Fifth Assessment Report of the Intergovernmental Panel on Climate Change* (eds. Stocker, T. F. *et al.*) 1137–1216 (Cambridge University Press, 2013).
 50. Bamber, J. L. & Aspinall, W. P. An expert judgement assessment of future sea level rise from the ice sheets. *Nat. Clim. Chang.* **3**, 424–427 (2013).
 51. Gomez, N., Mitrovica, J. X., Huybers, P. & Clark, P. U. Sea level as a stabilizing factor for marine-ice-sheet grounding lines. *Nat. Geosci.* **3**, 850–853 (2010).
 52. Kingslake, J. *et al.* Extensive retreat and re-advance of the West Antarctic Ice Sheet during the Holocene. *Nature* **558**, 430–434 (2018).
 53. de Boer, B., Stocchi, P., Whitehouse, P. L. & van de Wal, R. S. W. Current state and future perspectives on coupled ice-sheet – sea-level modelling. *Quat. Sci. Rev.* **169**, 13–28 (2017).
 54. DeConto, R. M. & Pollard, D. Contribution of Antarctica to past and future sea-level rise. *Nature* **531**, 591–597 (2016).

Reviewers' comments:

Reviewer #4 (Remarks to the Author):

In general, I think the authors have replied well to comments from all reviews. There's quite some discussion on several points in the manuscript, but overall these are now noted in the text and/or supplement.

Concerning the reviews (I have not read all in full detail), I think there's quite some work to be done to get a coherent picture of all LIG data, and the separate contributions of the GrIS and AIS, and possible other factors that impact regional sea level changes. This paper will certainly add to that discussion, which is by far not done yet.

Reviewer #5 (Remarks to the Author):

This paper presents an intriguing interpretation of Last Interglacial sea level and ice sheet contributions. This paper comes to me after having already been reviewed by several people and with a rebuttal from the authors addressing the comments of 4 reviewers.

I think the argument presented by the authors is very interesting and may be largely correct – in that the Antarctic and Greenland ice sheets may have had their peak contributions to LIG sea level asynchronously—which is largely evident already from the existing literature. There are several issues that prevent me from supporting publication of the manuscript in its current form, as follows:

(1) The authors emphatically state in the abstract that this paper presents “new data” in the abstract but do not specify what kind of data, nor can I find any indication of new data anywhere in the text, methods, or supplement, nor any reference to a new data table. The only thing here I see that is new is a new age model for the Red Sea RSL reconstruction across the LIG, which appears to be a function of moving a single tie-point to stretch the curve. To that end, this is more of a new interpretation of existing data, but it seems disingenuous at best to make a claim for new data when it is not transparent at all that this is true.

(2) Many of the key claims and numerical calculations rest on understanding the uncertainty of the proxy data that they are using. The authors appear to be rigorous in their discussion of the uncertainties and in propagating them through, but then report the rates of sea-level change – which feature prominently in the discussion section – with no uncertainties attached to them! What are the uncertainties here, are they larger than the rates themselves? This is critically important in the interpretation and assessing the significance of the values they present.

(3) Also with respect to uncertainty, they demonstrate in the supplement a variety of GIA models and then apply a GIA correction, but again, it does not appear that they have attached any uncertainty to the GIA correction from my reading of the supplement. This correction is pivotal. It can change the total sense of direction of the evolution of GMSL over the LIG based on what the GIA prediction is for the Hanish Sill. If they have chosen the wrong curve, then this can totally change the timing and magnitude of AIS contribution as they calculate. Can they address this and perhaps present two different scenarios on how it changes the calculation? Though their predicted RSL curves are quite flat for the ICE-3 and -4 scenarios that they prefer, this disagrees with the prediction of Lambeck et al., 2011, QSR, who also used a larger Eurasian ice sheet in his ice model and showed 4-5 m drop in RSL (with no excess meltwater) during the LIG at sites near the sill. The larger point being that there is significant disagreement between models about what this GIA correction is. It seems to me that this correction really drives the conclusions about when and how much the AIS contributed to GMSL. Given that, it is not clear that they can claim to know when and how much the AIS contributed, given the existing uncertainties.

(4) Fundamentally, this is yet another age-model revision of the Red Sea RSL record, of which there have been many. The movement of the tie point (from 123 ka to 118.5 ka) shifts many ages outside of their previously reported uncertainty (from earlier papers), and although the authors cite a 1.2 ky uncertainty, based on past revisions to this age model, why should one believe this statement to be true given previous adjustments to the age model? The authors choose a 118.5 ka

tie point, but based on the available data, it seems that tie point could just as easily be 116 or 117 (+/- 1.2) or 119 ka... The justification for why this particular number was chosen was lacking. Because the discussion centers so much on the rates, the uncertainties here are important. I'm not really sure how the authors can address this except not to emphasize the rates – or timing.

Other considerations:

--- The discussion of the coral data is extensive in the supplement, but lacking a summary there of what is seen. It seems like some basic summary statements could help here, e.g., it seems clear that the majority (all?) of the sites record multiple phases/generations of reef growth, that existing dates show a progression of younger reefs on above, on top of older reefs, are the reefs in direct superposition at all the sites (except the tectonic uplifting ones), etc.. You have emphasized that there are differences, but stating the commonalities and extracting what you learn from this comparison would be more useful and would justify the very long and detailed description in the supplement. As is, there is a detailed supplement, and then some statements are made in the main text that were not emphasized in the supplement, so it is not clear how one supports the other.

--- In the conclusion section, the authors provide some discussion of possible reasons for such high rates, but none of these point to the Red Sea as part of the explanation. For example, is it possible that these oscillations are amplified by climate feedbacks that affect the evaporation/precipitation balance in this region?

--- Why is a different Red Sea curve (polynomial curve) used in Supplemental figure S3 rather than the core that is mostly discussed in the text?

--- Also in Supp Fig S3, why is there a data point from the Seychelles near the 118.5 ka tie point? I thought the published Seychelles record ended earlier.

--- I didn't see any table with the new chronology for the Red Sea record, which would have to be included if published.

Response to reviewers: *Rohling et al.*, "Asynchronous Antarctic and Greenland ice-volume contributions to the last interglacial sea-level highstand" *N. Comms*.

We thank the reviewers for their thorough and useful comments. Our responses to specific comments are given below, with actions highlighted in yellow.

Reviewer #4

Thank you. The revised manuscript ("replied well to comments from all reviews") benefited from your useful and insightful comments.

Reviewer #5

1. Perceived lack of new data:

We do not understand this comment. New data is presented, as is clearly stated in the *Methods* section (line 460). This is a high-resolution Last Interglacial sea-level data from a northern Red Sea (core KL23), which provides essential validation of the key signal used in the study. Also, 2/3 of the $\delta^{18}\text{O}_{\text{sw}}$ record is new in that it has not yet been published (it has been on a few posters).

We will provide full data on acceptance, for all of our records shown in the manuscript, including the probabilistic analysis.

2. Rates of sea-level change and their uncertainties:

A strength of the Red Sea record is that it is a continuous, and high-resolution record, unlike many other Last Interglacial (LIG) sea-level records which have comparatively sparse temporal coverage. We admit that we gave only rate of change information for median and modal records, and used comparison between those as range estimates.

To address the reviewer's concerns, we have now added fuller expressions of the uncertainties in the main text discussion.

3. Disagreement between different GIA corrections and uncertainties:

The full uncertainties associated with a GIA correction encompass both parametric and structural components. Some of these are unknown, and others poorly characterised, and this is an active topic of ongoing research. For the Red Sea, Lambeck *et al.* (2011)^{ref.1} use a region-specific radially symmetric approach, which can be tuned to regional conditions such as mantle viscosity. Our settings/reconstructions include similar values to Lambeck's parameters, yet investigate also the impact of different ice distribution/volume, which Lambeck *et al.* did not do to the same extent. However, as with Lambeck, we do not have any conclusive arguments to prefer any of the Earth models EM1, EM2, or EM3 (note, for illustration only EM4 is deliberately extreme, and is too far outside Lambeck's and Peltier's values to be realistic).

However, the reviewer is correct that we forgot to propagate uncertainties in the GIA corrections from *Supplementary Information*, section 4. We have now included those uncertainty bounds in Figure 3. The GIA correction including the uncertainties across EM1, 2 and 3 then runs from 0 ± 0 m at 135 ka, toward 4 ± 2 m at 115 ka (we use a linear adjustment in between, as before). Many thanks for pointing out this omission.

4. Red Sea age model:

We disagree with the reviewer that at its "fundamental" level, the paper rests on a refinement of the age model for the Red Sea. Instead, the paper presents:

- (1) new data, both in validation of the Red Sea record and for the Greenland melt contribution (through a newly applied multi-endmember mixing model, which one of us had pioneered for modern condition in 2010),
- (2) a newly quantified deconvolution of the contributions of the polar ice sheets during the LIG, and
- (3) a wealth of supporting palaeoceanographic evidence, and (in the Supplement) reef evidence.

We object to the statement of "many" revisions of the Red Sea chronology. In fact, this paper concerns only the second revision to the original age model presented in Siddall *et al.*, 2003^{ref.2}, which was extended to the full 500 ka in Rohling *et al.* (2009)^{ref.3}. The first revision was presented in Grant *et al.* (2012)^{ref.4}, and this methodology was subsequently also extended (but not changed in the last 150 kyr) to the full 500 ka

record in Grant *et al.* (2014)^{ref.5}. The age model presented here is therefore only the second revision. In all cases, revision came about from an advent of new information that was not previously available. Revisions to age models are an accepted (and necessary) practise within palaeo studies, resulting from increased data and/or understanding of the Earth system and its interactions. Most ice-core chronologies have undergone several age model revisions. For example, EPICA Dome C: EDC₁ (Scwander *et al.*, 2001^{ref.6}); EDC₂ (EPICA community members, 2004^{ref.7}); EDC₃ (Parrenin *et al.*, 2007^{ref.8}); synchronisation between Dome-C and Dronning Maud Land, EDML₁ (Ruth *et al.*, 2007^{ref.9}), improvements in volcanic synchronisation between Dome-C and Vostok (Parrenin *et al.*, 2012^{ref.10}), and so on. The same is true *in extenso* for orbitally tuned chronologies and radiocarbon calibration studies. We fail to see how chronological improvements can be seen in any other way than as a symptom of progression of the science.

The 1.2 ka age uncertainty (at 95%) derives from Grant *et al.* (2012, 2014^{refs.4,5}), and is not adjusted here for the sections where the age model is kept constant relative to that age model. When performing the probabilistic assessment for uncertainty propagation, we use the newly diagnosed uncertainties from *Supplementary Figure 2*. These are larger (therefore, more conservative) than the Grant *et al.* uncertainties in the interval 120-110 ka (Suppl. Fig. 2).

This was perhaps not too clearly stated; we have added substantial text to the *Methods* to clarify this.

Choice of 118.5 ka: As stated in the main manuscript (lines 102 to 103), *Methods* (lines 445 to 448) and *Supplementary Information*, section 2 (p13), this age derives from the radiometrically dated Yucatan Peninsula speleothem record (Moseley *et al.*, 2013^{ref.11}) which indicates that sea level must have dropped below 0 m at this time. This interpretation is supported by coral data from Barbados (Cutler *et al.*, 2003^{ref.12}) and other far-field sites (Hibbert *et al.*, 2016)¹³. We therefore imposed the Yucatan speleothem temporal and elevation constraint on the Red Sea record, such that there is only a marginal, 2.5 % chance that the Red Sea record exceeds the Yucatan record at 118.5 ka (i.e., the upper bound of the 95% confidence interval sits at this elevation at 118.5 ka).

This may not have been sufficiently clear; we have added a sentence to the *Methods* section in the main manuscript to clarify this.

Also, we emphasize that – in validation – Figure 2f shows the KL11 probabilistic curve alone, and it is evident that also its 95% probability zone for individual datapoints (light grey) drops below 0 m at the same 118.5 ka. We do not use this as an argument, but only in validation, to avoid circularity.

5. Coral data – lack of summary in the *Supplementary Information*:

The format of the section 1 of the *Supplementary Information* is a direct consequence of previous reviews. We did initially synthesise the emerging consensus from (mainly) coral records, especially sites that had been intensively sampled, with good stratigraphic control, and well-dated, in both the *Supplementary Information* and as a plot in figure 2. However, original reviewer 2 was adamant that this synthesis should be removed, and we did so accordingly. But a moderated synthesis still exists in the Suppl. Info section 1. The current reviewer may have missed the previous (fierce) debate with reviewer 2, and thus may be expecting something that we also liked to include, but we were told previously to leave this out in no uncertain terms.

6. Red Sea – potential for the amplification of the sea-level signal through climate feedbacks:

This is a misconception. The Red Sea method is based upon the sea level control of the residence time of sea water within a highly evaporative semi-enclosed basin. The various influences on $\delta^{18}\text{O}_{\text{calcite}}$ are well constrained (Siddall *et al.*, 2003, 2004^{refs.2,14}) and fully included in the sea-level model's uncertainty through sensitivity testing (Siddall *et al.*, 2004)^{ref.14}. That study demonstrates that the effect of changes in sea level far exceeds changes in other factors such as evaporation and temperature. Specifically, temperature increase INcreases sea-water $\delta^{18}\text{O}$ but DEcreases calcite $\delta^{18}\text{O}$ through water-to-calcite fractionation; both processes are fully accounted for in the model and its sensitivity tests. Finally, the model uncertainty bounds span conditions in which the basin is "locked" year-round in winter conditions, and year-round in summer conditions; effectively, this gives $\pm 50\%$ allowances in temperature and evaporation uncertainty. This exceeds even reconstructions of glacial-interglacial contrasts in the area, let alone centennial variations. The fact that the rates are found in signals that exceed the method's uncertainty bounds (after

probabilistic evaluation) indicates that they *exceed* any climatic influences. From Rohling *et al.* (2009)^{ref.3}, we furthermore highlight the passage (from Methods) copied below in evidence of the sea-level dominance on the Red Sea signal. Finally, studies on runoff-biomarkers and foraminiferal distribution patterns have not indicated any major impacts through the Holocene and LIG, except for some "*negligible term[s] in the overall Red Sea freshwater budget*" (Trommer *et al.*, 2010, 2011^{refs.15,16}). In short, not a single conclusive study has been presented since the Red Sea method was first published 2 decades ago (Rohling *et al.*, 1998^{ref.17}, Siddall *et al.*, 2003^{ref.2}) about any significant temperature or evaporation complications on the method's sea-level determinations. On the contrary, the method's analytical solution and uncertainties have been comprehensively validated with an independent, numerical approach (Biton *et al.*, 2008^{ref.18}).

All of this has been documented extensively in many papers before, so that we do not feel it necessary to keep repeating it. The core of the problem seems to be a lack of wanting to engage with the quantitative nature of the method (some aspects are not easy to grasp intuitively because of opposing effects), rather than a lack of explanation in the literature of what goes into its uncertainties. **Accordingly, we don't think any response is needed to this comment.**

From Methods in Rohling *et al.* (2009)^{ref.3}:

*The new KLog data comprise three sets of stable oxygen isotope ($\delta^{18}O$) analyses: first... $\delta^{18}O$.. of dried, homogenized bulk sediment...; second, ... $\delta^{18}O$ of non-size-constrained picks of the planktonic foraminiferal species *G. ruber* from an independently obtained set of pilot samples...; and third, ... $\delta^{18}O$ analyses of 20–30 specimens of *G. ruber* (white) in a narrow (320–350 μm) size range, at high resolution (down to 150–300 yr)... The various analyses are highly reproducible (Fig. 1), even though the carrier media are different and (virtually) independent.... The high-resolution $\delta^{18}O_{\text{ruber}}$ analyses concern specific morphotypes of a particular planktonic foraminiferal species that lives in the uppermost 50 m of the water column.... The $\delta^{18}O_{\text{bulk}}$ analyses represent an indiscriminate mix of carbonate materials from planktonic and benthic foraminifera, nanofossils, pteropods, halimeda, inorganically precipitated carbonates and so on (CaCO_3 concentrations between 60 and 90%). That the $\delta^{18}O$ of such vastly different carriers reveals the observed consistent variability (Fig. 1a) bears testimony to the common forcing of Red Sea $\delta^{18}O$, namely sea-level controlled changes in the residence-time of sea water in the basin....*

7. Supplementary Figure 3 – use of polynomial of entire record rather than KL11:

As mentioned above, Figure 2f shows the KL11 probabilistic curve, and it is evident that the 95% probability zone for individual datapoints (light grey) drops below 0 m at the same 118.5 ka. We do not use this as an argument, but only in validation, to avoid circularity. Given that this is dealt with in the main text, we don't see a need to duplicate it in the Supplement. Main-text figure 2 shows clearly the temporal relationship between the whole-stack polynomial and the KL11-only probabilistic assessment.

We have added a sentence to clarify this matter to the caption of Suppl. Fig. S3.

8. Seychelles data point at ~ 118.5 ka tie-point:

The reviewer has made a good spot here, and we might have explained our arguments behind this single point. **The argumentation has been added to the caption of Suppl. Fig. S3.**

9. Table with new chronology:

All data will be provided on acceptance, including the Red Sea record on the new chronology (see query 1).

References cited:

1. Lambeck, K. *et al.* Sea level and shoreline reconstructions for the Red Sea: isostatic and tectonic considerations and implications for hominin migration out of Africa. *Quat. Sci. Rev.* **30**, 3542–3574 (2011).
2. Siddall, M. *et al.* Sea-level fluctuations during the last glacial cycle. *Nature* **423**, 853–858 (2003).
3. Rohling, E. J. *et al.* Antarctic temperature and global sea level closely coupled over the past five glacial cycles. *Nat. Geosci.* **2**, 500–504 (2009).
4. Grant, K. M. *et al.* Rapid coupling between ice volume and polar temperature over the past 150,000 years. *Nature* **491**, 744–747 (2012).
5. Grant, K. M. *et al.* Sea-level variability over five glacial cycles. *Nat. Commun.* **5**, 5076 (2014).
6. Schwander, J. *et al.* A tentative chronology for the EPICA Dome Concordia Ice Core. *Geophys. Res. Lett.* **28**, 4243–4246 (2001).
7. Augustin, L. *et al.* Eight glacial cycles from an Antarctic ice core. *Nature* **429**, 623–628 (2004).

8. Parrenin, F. *et al.* The EDC₃ chronology for the EPICA Dome C ice core. *Clim. Past* **3**, 485–497 (2007).
9. Ruth, U. *et al.* 'EDML1': a chronology for the EPICA deep ice core from Dronning Maud Land, Antarctica, over the last 150 000 years. *Clim. Past* **3**, 475–484 (2007).
10. Parrenin, F. *et al.* Volcanic synchronisation between the EPICA Dome C and Vostok ice cores (Antarctica) 0–145 kyr BP. *Clim. Past* **8**, 1031–1045 (2012).
11. Moseley, G. E., Smart, P. L., Richards, D. A. & Hoffmann, D. L. Speleothem constraints on marine isotope stage (MIS) 5 relative sea levels, Yucatan Peninsula, Mexico. *J. Quat. Sci.* **28**, 293–300 (2013).
12. Cutler, K. B. *et al.* Rapid sea-level fall and deep-ocean temperature change since the last interglacial period. *Earth Planet. Sci. Lett.* **206**, 253–271 (2003).
13. Hibbert, F. D. *et al.* Coral indicators of past sea-level change: A global repository of U-series dated benchmarks. *Quat. Sci. Rev.* **145**, 1–56 (2016).
14. Siddall, M. *et al.* Understanding the Red Sea response to sea level. *Earth Planet. Sci. Lett.* **225**, 421–434 (2004).
15. Trommer, G. *et al.* Millennial-scale variability in Red Sea circulation in response to Holocene insolation forcing. *Paleoceanography* **25**, (2010).
16. Trommer, G. *et al.* Sensitivity of Red Sea circulation to sea level and insolation forcing during the last interglacial. *Clim. Past* **7**, 941–955 (2011).
17. Rohling, E. J. *et al.* Magnitudes of sea-level lowstands of the past 500,000 years. *Nature* **394**, 162–165 (1998).
18. Biton, E., Gildor, H. & Peltier, W. R. Red Sea during the Last Glacial Maximum: Implications for sea level reconstruction. *Paleoceanography* **23**, PA1214 (2008).
19. Dutton, A., Webster, J. M., Zwartz, D., Lambeck, K. & Wohlfarth, B. Tropical tales of polar ice: evidence of Last Interglacial polar ice sheet retreat recorded by fossil reefs of the granitic Seychelles islands. *Quat. Sci. Rev.* **107**, 182–196 (2015).

REVIEWERS' COMMENTS:

Reviewer #5 (Remarks to the Author):

(1) Regarding the comment on "where is the new data?", thank you for response. This is much clearer to me now, but is really buried in the manuscript. For example you say there is new $\delta^{18}O$ data, but in the Methods it is described like this:

Our Eirik Drift (core MD03-2664) surface sea-water $\delta^{18}O$ record ($\delta^{18}O_{sw}$) was
510 determined using the palaeotemperature equation of Ref. [70], with a Vienna
511 PeeDee Belemnite to Standard Mean Ocean Water standards conversion of
512 0.27‰, using $\delta^{18}O$ [71] and Mg/Ca temperature data[72] for the planktonic
513 foraminiferal species *Neogloboquadrina pachyderma* (sinistral), on the
514 chronology of Ref. [72].

Where you provide a reference for the $\delta^{18}O$ data, hence I thought it all came from that paper. Please reword this sentence to make it clear that there is also new data, maybe list number of data points or something? It is common to add info on instrumentation etc. also, but this is missing as well.

Also, I am now clear on the new Red Sea data, but again, can this be made clearer earlier in the paper?

Related to this, I am frankly appalled that the editors have seen it fit to send a paper for review that does not include the data. This is the first time in all my years reviewing papers that I have been asked to review a paper where the data has been withheld. Clearly this created confusion in the reviewing process as seen by this back-and-forth and it does not allow the paper to be properly reviewed. This greatly undermines the peer review process. While I have some trust that these particular authors have collected said data, this makes me uncomfortable. Are we now expected to accept papers for publication without anyone ever first seeing the data??! Imagine the problems that this could create.

(2) Uncertainties for rates of sea-level change:
Thank you for including these now.

(3) Uncertainties for GIA correction:
This is improved, but states that there is zero uncertainty at 135 ka. I agree with what the authors imply, that is, that what is more important here is the slope of the GIA correction rather than the absolute amount (starting at 135 ka) – at least, when talking about rates of change. A statement to this effect, that the uncertainty is not zero for the absolute position of sea level, but that there is some unknown uncertainty here, would be appropriate.

(4) Red Sea Age model.
Thank you. This was also largely clarified by your explanation of what was new data and what was not.

(5) Coral data summary. Thank you for your explanation. I had sensed there was some history in the review process here.

(8) Seychelles data point. Thank you for clarifying. Not important to the paper, but it is strange that you would use a screening process that allows the two most altered compositions for this coral to screen through – or to use data from a coral at all that shows such heterogeneous compositions. The average of two altered data points does not make a better data point.

(9) Tables. See my comment above in #1. This is highly irregular. In any case, at a minimum the data must be included in a manner that is usable by others to be published.

Response to reviewers: *Rohling et al.*, "Asynchronous Antarctic and Greenland ice-volume contributions to the last interglacial sea-level highstand" *N. Comms*.

We thank the reviewer for reassessing the manuscript. Our responses to comments requiring further clarification are given below, with actions highlighted in yellow.

Reviewer #5

1. New data:

We have amended the manuscript in line with your suggestions. Analytical details for the new Red Sea $\delta^{18}\text{O}$ and relative sea level, as well as the Eirik Drift $\delta^{18}\text{O}_{\text{sw}}$ data is given in lines 370 to 379 and 458 to 465, respectively. The new data and supporting data is also contained with the accompanying Source Data file (<http://doi.org/10.6084/m9.figshare.9790844>).

2. Uncertainties for rates of sea-level change:

Addressed in previous response (retained in the revised version of the manuscript).

3. Uncertainty in GIA correction:

We have added some clarifying text to the *Supplementary Information (Supplementary Note 4, page 19 of the supplement)*.

4. Red Sea age model:

Addressed in previous response.

5. Coral summary:

Addressed in previous response.

8. Seychelles data point at ~ 118.5 ka tie-point:

We selected commonly used age reliability screening criteria (e.g., Dutton and Lambeck, 2012^{ref.1}) but as the signature of carbonate diagenesis may be subtle, as rightly pointed out by the reviewer, screening data will successfully remove some, but not all, of the altered samples.

We also agree that an average of the replicate age determination leaves much to be desired, however, this is what is still commonly done within the community (e.g., Dutton, 2015^{ref.2}, Yokoyama et al., 2018^{ref.3}).

9. Data tables:

These have now been supplied as a Source Data file (<http://doi.org/10.6084/m9.figshare.9790844>). This data file will also be lodged on <http://www.highstand.org>.

References cited:

1. Dutton, A. & Lambeck, K. Ice volume and sea level during the last interglacial. *Science* (80-.). **337**, 216–219 (2012).
2. Dutton, A. Uranium-thorium dating. in *Handbook of Sea-Level Research* (eds. Shennan, I., Long, A. J. & Horton, B. P.) 386 (John Wiley and Sons Ltd., 2015).
3. Yokoyama, Y. *et al.* Rapid glaciation and a two-step sea level plunge into the Last Glacial Maximum. *Nature* **559**, 603–607 (2018).